- Rapid Secondary Organic Aerosol Formation at the Air–Water Interface from
- Methoxyphenols in Wildfire Emissions: UVA-Driven S(IV) Photooxidation
- to Organosulfates
- Authors: Baohua Cai, <sup>1</sup> Yuanlong Huang, <sup>2</sup> Wenqing Jiang, <sup>3</sup> Yanchen Li, <sup>1</sup> Yali Li, <sup>1</sup> Jinghao Zhai, <sup>1,4</sup> Yaling
- Zeng, <sup>1,4</sup> Jianhuai Ye, <sup>1,4</sup> Huizhong Shen, <sup>1,4</sup> Chen Wang, <sup>1,4</sup> Lei Zhu, <sup>1,4</sup> Tzung-May Fu, <sup>1,4</sup> Qi Zhang, <sup>3\*</sup> Xin
- Yang<sup>1,4</sup>\*
- <sup>1</sup>Shenzhen Key Laboratory of Precision Measurement and Early Warning Technology for Urban
- Environmental Health Risks, School of Environmental Science and Engineering, Southern University of
- Science and Technology, Shenzhen 518055, China.
- <sup>2</sup>Ningbo Institute of Digital Twin, Eastern Institute of Technology, Ningbo 315201, China.
- 3Department of Environmental Toxicology, University of California, Davis, California 95616, United States.
- <sup>4</sup>Guangdong Provincial Field Observation and Research Station for Coastal Atmosphere and Climate of the
- Greater Bay Area, Southern University of Science and Technology, Shenzhen, Guangdong, 518055, China
- \*Qi Zhang and Xin Yang.
- Email: yangx@sustech.edu.cn and dkwzhang@ucdavis.edu
- Keywords: Photooxidation, Atmospheric Organosulfates, Sulfate.
- This PDF file includes:
- Main Text
  Figures 1 to 6
  Table 1

Abstract

Wildfire emissions release large amounts of methoxyphenols, which serve as key precursors of aqueous-phase secondary organic aerosols (SOA). Their transformation is closely coupled with aqueous S(IV) oxidation, jointly driving the formation of sulfate and organosulfates; however, the underlying mechanisms remain poorly understood. Here, we identify a novel, metal-free mechanism for  $SO_4^{\bullet-}$  generation under UVA light (370 nm), supported by experiments and quantum chemical calculations. Photolysis of the  $[SO_3^{2-}+O_2]$  complex yields a  $[SO_3^{\bullet-}+O_2^{\bullet-}]$  pair that forms peroxomonosulfate  $(SO_5^{\bullet-})$  and ultimately  $SO_4^{\bullet-}$ . These radicals rapidly oxidize guaiacol, a biomass burning phenol, in bulk solution ( $k = 3.8 \times 10^{10} \text{ M}^{-1} \text{ s}^{-1}$ ), producing SOA enriched in organosulfates. Microdroplet experiments show 100-fold rate enhancement due to interfacial effects. Box and global modeling indicate that this aqueous UVA pathway is a significant, previously overlooked source of sulfate. This work established a new photochemical link between S(IV) oxidation and SOA formation, with implications for aerosol composition, oxidative capacity, and climate-relevant processes.

## 37 Abstract Graphic

1 INTRODUCTION

65

75

Wildfires are occurring with increasing frequency, intensifying climate perturbation and exacerbating human health risks (Zhao et al., 2025; Teymoor Seydi et al., 2025). Their emissions, rich in methoxyphenols and other semi-volatile organic compounds, readily partition into cloud and aerosol water, where they undergo rapid transformations that produce substantial amounts of SOA (He et al., 2024; Li et al., 2023a; Liu et al., 2022). These aqueous-phase reactions are not isolated; rather, they are intricately coupled with other atmospheric chemical processes, resulting in complex multiphase chemistry that remains poorly understood. Sulfate is a major component of fine particulate matter (PM), with significant impacts on air quality and public health (Wang et al., 2016; Abbatt et al., 2006). In the atmosphere, sulfate forms primarily through gasphase SO<sub>2</sub> oxidation by hydroxyl radicals (•OH) and aqueous-phase oxidation of S(IV) species, such as dissolved SO<sub>2</sub>, HSO<sub>3</sub>, and SO<sub>3</sub><sup>2</sup>, in cloud, fog, or aerosol water. The aqueous-phase pathway includes direct oxidation by H<sub>2</sub>O<sub>2</sub> (Liu et al., 2020), O<sub>3</sub> (Hoffmann, 1986; Lan et al., 2011), and NO<sub>2</sub> (Zhang and Chan, 2023; Gao et al., 2022; Liu and Abbatt, 2021); catalytic oxidation mediated by transition metal ions (e.g., Fe and Mn) (Zuo et al., 2005; Wang et al., 2021; Harris et al., 2013; Brandt and Eldik, 1995); and photocatalytic processes involving humic-like substances (HULIS) in the presence of O2 (Wang et al., 2024; Pan et al., 2024). Despite extensive research, substantial discrepancies remain between observed sulfate levels and model predictions, indicating missing or poorly characterized pathways (Zheng et al., 2015). Beyond sulfate formation, aqueous S(IV) oxidation can also form organosulfates (OSs) in the presence of volatile organic compounds (VOCs) (Passananti et al., 2016; Duporté et al., 2020; Surratt et al., 2008; Iinuma et al., 2007; Darer et al., 2011; Riva et al., 2015). OSs constitute a substantial fraction (e.g., 5-30%) of the organic mass in atmospheric PM (Shakya and Peltier, 2015; Tolocka, 2012; Hughes et al., 2021; Romero and Oehme, 2005) and provide an important chemical link between sulfur and organic aerosols. As amphiphilic molecules, OSs affect aerosol surface activity and hygroscopicity (Riva et al., 2019), thereby enhancing their potential to act as cloud condensation nuclei (CCN) (Peng et al., 2021). Some OSs are also linked to adverse health outcomes, including oxidative stress and proinflammatory responses in human lung cells (Khan et al., 2023). The sulfate radical (SO<sub>4</sub>) is a highly reactive intermediate in aqueous-phase S(IV) oxidation (Rudzinski et al., 2009), capable of rapidly oxidizing a wide variety of VOCs, including aldehydes (Coddens et al., 2018; Tran et al., 2022), olefins (Schindelka et al., 2013; Ren et al., 2021), phenols (Cope et al., 2022), and polycyclic aromatic hydrocarbons (Wang et al., 2008). These reactions produce oxidized organics that can subsequently form OSs through acid-catalyzed esterification or radical termination reactions. Solar radiation is a key driver of such radical chemistry, including the generation of SO<sub>4</sub>. (George et al., 2015; Herrmann et al., 2015). For example, in high-ionic-strength aerosol solutions (e.g., 3.7 M ammonium sulfate), SO<sub>4</sub> forms under UVB (~ 310 nm) irradiation, reaching steady-state concentrations near 10<sup>-12</sup> M (Cope et al., 2022). While direct photolysis of S(IV) species by UVC radiation can also yield SO<sub>4</sub>. (Cao et al., 2021), UVC is largely absorbed by the stratosphere and thus negligible in the troposphere.

Although UVA radiation is the dominant solar band at the Earth's surface, its role in aqueous S(IV) oxidation remains poorly understood. A recent study suggests that UVA light can promote SO<sub>2</sub> oxidation at the air-water interface (Gong et al., 2022), but the mechanisms and broader implications are unclear. To address this gap, we combined laboratory experiments with quantum chemical calculations to investigate a novel, metal-free UVA-induced pathway for SO<sub>4</sub>. generation. Using guaiacol (GUA), a representative biomass burning phenol, as a molecular probe, we tracked radical activity and OSs formation. We also explored how droplet microphysics and interfacial effects enhance this chemistry. Our findings reveal a previously overlooked UVA-driven mechanism for sulfate and OSs formation, with important implications for atmospheric chemistry, air quality, and climate.

86 87

88

## 2 MATERIALS AND METHODS

#### 2.1 Materials

- Guaiacol (99%), sodium sulfite (Na<sub>2</sub>SO<sub>3</sub>, >99%), 2,2,6,6-tetramethyl-1-piperinedinyloxy (TEMPO, 98%),
- ethanol (99%), and tert-butanol (99%) were purchased from Macklin. Zero air is made up of 21% O<sub>2</sub> and 79%
- N<sub>2</sub>. All water used in the experiments was ultrapure Milli-O water (18.2 M $\Omega$  cm<sup>-1</sup>).

93

#### 2.2 Experimental Methods

- Bulk aqueous experiment. All experiments were performed in a 25 mL airtight Pyrex tube equipped with a
- magnetic stir bar and a gas inlet tube for feeding high-purity zero air or nitrogen under 370 nm light or LED
- light irradiation. A 20 mL reaction solution containing guaiacol, Na<sub>2</sub>SO<sub>3</sub>, and other reactants was prepared.
- The pH of the reaction solution was adjusted using H<sub>2</sub>SO<sub>4</sub> and NaOH and measured with a pH meter REDOX
- potentiometer Conductivity meter (AZ-86555) that was calibrated with commercial pH standards. In
- experiments requiring the measurement of total inorganic sulfur, the pH is adjusted with either phosphoric
- acid or phosphate. Aliquots (3 mL) were sampled every 20 min for 1h, with 0.30 mL MeOH added
- immediately to quench the reaction. Each experiment was repeated at least twice.
- HPLC analysis. The concentrations of the guaiacol and phenol were determined using an HPLC (Thermo
- Scientific<sup>TM</sup> UltiMate<sup>TM</sup> 3000) equipped with a diode array detector (DAD) and an Agilent 5 TC-C18 column
- (150×4.60 mm, 5 μm). The column temperature was maintained at 25 °C, and the flow rate was set to 1
- 105 ml/min. Detection was performed at 274 nm. The mobile phase consisted of 60/40 (v/v) acetonitrile/water
- acidified with 0.1% trifluoroacetic acid (TFA).
- Direct infusion HRMS. Reaction solutions were filtered through a membrane and then directly introduced
- into an Agilent 6546 quadrupole time-of-flight mass spectrometer (QTOF-MS, Santa Clara, CA) with
- electrospray ionization (ESI) source in negative mode. The MS parameters were as follows: nebulizer, 25
- psi; gas flow, 10 L/min; sheath gas temperature, 330 °C; capillary voltage, 3500 V; sheath gas flow, 12 L/min.
- MS data were collected in an m/z range of 90-500. Agilent MassHunter Qualitative Analysis software
- (version 10.0) was used for data analysis.

https://doi.org/10.5194/egusphere-2025-5323 Preprint. Discussion started: 12 November 2025 © Author(s) 2025. CC BY 4.0 License.

to monitor guaiacol absorbance during reaction with Na<sub>2</sub>SO<sub>3</sub> and to record sample spectra from 200 to 500 115 nm. The reaction solution was directly loaded without dilution or modification. Spectra of the guaiacol -116 Na<sub>2</sub>SO<sub>3</sub> reaction were collected every 20 min for 1 h. 117 IC measurements. Sulfite (SO<sub>3</sub><sup>2</sup>) and sulfate (SO<sub>4</sub><sup>2</sup>) concentrations were analyzed using a Metrohm 883 118 Basic IC system quipped with a Metrosep A supply 5-250/4.0 analytical column and a conductivity detector 119 (Liu et al., 2025). Prior to analysis, 2% isopropanol and 1.0 mM NaOH was added into the reaction solution. 120 The eluent used was 3.2 mM Na<sub>2</sub>CO<sub>3</sub>/1.0 mM NaHCO<sub>3</sub>, with a flow rate of 0.8 mL min<sup>-1</sup>. For total inorganic 121 sulfur analysis, samples were pre-oxidized to  $SO_4^{2-}$  using hydrogen peroxide (H<sub>2</sub>O<sub>2</sub>) before IC detection. 122 FIDI-MS experiment. Droplets approximately 2 mm in diameter (~4 µL volume) were suspended from the 123 tip of a stainless-steel capillary, which was positioned equidistantly between two parallel plate electrodes 124 separated by 6.3 mm apart. The droplets were formed by injecting the analyte solution through the capillary 125 using a syringe pump. The parallel plates were mounted on a translation stage to align the front electrode's 126 aperture with the atmospheric pressure inlet of a Thermo-Fischer LTQ-XL mass spectrometer (Waltham, 127 MA), which was operated under laboratory ambient air. Once the droplets were formed, a 60-second 128 equilibration period was allowed to enable compound diffusion and achieve equilibrium coverage at the air-129 water interface. 130 Sampling of the suspended droplets was accomplished by applying a high-voltage pulse (3 - 5 kV, 100 ms 131 duration, variable polarity) to the rear electrode and capillary, with half the voltage simultaneously applied 132 to the rear plate, thereby establishing a uniform electric field. This field induced a dipole in the suspended 133 droplets, causing it to elongate and form a double Taylor cone at both ends, which ejected oppositely charged 134 submicron-sized droplets. These negatively charged droplets passed through the aperture of the front plate 135 and entered the mass spectrometer for gas-phase ion detection. Due to the significant disturbance caused by 136 the ionization droplet interface (IDI) sampling, a new droplet was generated for each measurement. In this 137 study, a negative voltage polarity was applied to the rear plate and capillary to facilitate detection of 138 deprotonated guaiacol ions ([GUA]-). 139 HR-ToF-AMS experiment. During photochemical experiments, reaction solutions were aerosolized with a 140 constant output atomizer (TSI Inc.) using N2 as the carrier gas. The resulting aerosols were dried with a 141 diffusion dryer and then introduced into a high-resolution time-of-flight aerosol mass spectrometer (HR-ToF-142 AMS; Aerodyne Research, Inc.) for chemical characterization. Drying allowed evaporation of volatile and 143 semi-volatile species; therefore, the AMS primarily measured the mass concentration and bulk composition 144 of the remaining low-volatility products. The operating principles of AMS have been described previously (Decarlo et al., 2006; Canagaratna et al., 2007). Briefly, the AMS analyzes non-refractory aerosols that 145 146 vaporize at ~600 °C under high vacuum via 70 eV electron impact ionization. In this study, the AMS was 147 operated in "V" ion optical mode (mass resolution ~3000) to acquire mass spectra up to m/z 422. The AMS

UV-vis spectroscopy. An ultraviolet-visible Spectrophotometer (Youke, T2602, Shanghai, China) was used

data were processed with the standard AMS toolkits SQUIRREL (v1.67) and PIKA (v1.27), available at http://cires.colorado.edu/jimenez-group/ToFAMSResources/ToFSoftware/.

149150151

#### 2.3 Theoretical Calculations

- DFT and TDDFT calculations. Geometry optimizations and frequency calculation for all molecular 153 structures (reactants, products, and transition states) were performed using the M06-2X (Zhao and Truhlar, 154 2007) functional with the ma-TZVP basis set (Zheng et al., 2010), employing the SMD solvation model 155 (Marenich et al., 2009) to simulate aqueous-phase effects in water, as implemented in the Gaussian 16 156 software package (Frisch et al., 2016). Optimized structures were verified by frequency computations to 157 confirm local minima (zero imaginary frequencies) or transition structures (single imaginary frequency). 158 Intrinsic reaction coordinate (IRC) calculations were performed to ensure that the first-order saddle points 159 found were true transition states (TS) connecting the reactants and the products. Single-point energy 160 calculations and solvation effects were evaluated at the CCSD(T)/aug-cc-pVTZ (Guo et al., 2018; Noga and 161 Bartlett, 1987) level using the SMD solvation model, with geometries optimized at M06-2X/ma-TZVP and 162 zero-point energy (ZPE) correction applied. The calculations were carried out using the ORCA 5.0.3 program 163 package (Neese, 2012). Multiwfn 3.8 (Lu and Chen, 2012) and Shermo 2.4 (Lu and Chen, 2021) were used 164 for further data analysis.
- GROMACS 4.5.5 (Hess et al., 2008). In a cubic box with periodic boundary conditions, the system consisted
  of 1000 SPC/E water molecules and one GUA molecule using the OPLS-AA force field. Electrostatics were
  treated with the particle-mesh Ewald (PME) method; van der Waals interactions were truncated at 10 Å. A

Classical MD calculations. Classical molecular dynamics (MD) calculations were performed using

- leap-frog integrator was used with a 2 fs timestep, and the trajectories were recorded every 10 steps.
- Umbrella Sampling: To determine the average volume for each system, 10 ns simulations were conducted
  in the NVT ensemble, where the temperature was set to 300 K using the V-rescale method. The potentials of
  mean force (PMF) were calculated using the Weighted Histogram Analysis Method (WHAM) calculations,
  which were performed in one additional 10 ps simulation with initial configurations from the preceding
- which were performed in one additional 10 ns simulation with initial configurations from the preceding simulations. The GUA moved in the z-dimension around their frozen positions under a harmonic restoring force. The force constant was set at 1×10<sup>3</sup> (kJ/mol/nm), and configurations were recorded every 0.5 ps.
- Visualization and trajectory analysis were implemented using VMD (Humphrey et al., 1996).

177178

165

## 2.4 Model Calculation

- Box model conditions. Under illumination from the Kessil PR160L-370nm lamp, the observed photooxidation rate constant ( $k_{\text{obs}}$ ) for S(IV) is described by the following equation for pH  $\geq$  4.0:
- $-d[S(IV)]/dt = k [H^+]^{-0.47}[O_2] [S(IV)], \text{ where } k = 6.6 \times 10^{-3}$
- The lamp's intensity in the 340 400 nm range was measured as 850 W m $^{-2}$  using a PL-MW2000 strong
- light optical power meter (Beijing Perfectlight Technology Co., Ltd.), compared 63.3 W m<sup>-2</sup> for solar AM0

- radiation in the same range (John H. Seinfeld and Pandis, 2016). Therefore, the S(IV) photooxidation rate
- under sunlight is approximately 7.4% of that under the Kessil PR160L-370nm lamp. Sulfate production rates
- at 271 K were calculated for different aqueous-phase reaction pathways with O<sub>3</sub>, H<sub>2</sub>O<sub>2</sub>, TMIs, and NO<sub>2</sub>,
- following Cheng (Cheng et al., 2016), excluding ionic strength effects.
- The Henry's law constants at 271 K for  $SO_2$ ,  $O_3$ ,  $H_2O_2$ , and  $NO_2$  are 3.521 M/atm, 0.025 M/atm,  $1.147 \times 10^{-2}$
- $10^6$  M/atm, and  $2.319 \times 10^{-2}$  M/atm, respectively. Equilibrium constants for  $SO_2 \cdot H_2O$  are  $K_{S1} = 0.025$  M and
- $K_{S2} = 1.09 \times 10^{-7} \text{ M}$  (Cheng et al., 2016).
- Scenario Conditions. "Cloud droplets" scenario:  $[SO_2(g)] = 5$  ppb,  $[NO_2(g)] = 1$  ppb,  $[H_2O_2(g)] = 1$  ppb,
- $[O_3(g)] = 50 \text{ ppb}$ ,  $[Fe(III)] = 0.3 \mu\text{M}$ ,  $[Mn(II)] = 0.03 \mu\text{M}$ , liquid water content (LWC) =  $0.1 \text{ g/m}^3$ .
- "Beijing haze" scenario:  $[SO_2(g)] = 40$  ppb,  $[NO_2(g)] = 66$  ppb,  $[H_2O_2(g)] = 0.01$  ppb,  $[O_3(g)] = 1$  ppb,
- LWC =  $300 \mu g/m^3$ . The concentrations of Fe(III) and Mn(II) were assumed to vary with pH (Cheng et al.,
- **195** 2016).
- Globel seasonal simulation. Using global SO<sub>2</sub> concentrations (Fig. S24), cloud pH (Fig. S25), global
- aqueous-phase S(IV) concentration (Fig. S26), liquid water content (Fig. S27), and UVA radiation (345 -
- 412.5 nm; Fig. S28), we estimated global aqueous-phase sulfate production rates.
- The aqueous-phase S(IV) concentration was calculated as:

$$[S(IV)] (M) = [SO_2]_g (ppb) \times 10^{-9} \times (1 + \frac{K_{s1}}{[H^+]} + \frac{K_{s1} \times K_{s2}}{[H^+]^2}) \times H_{so2}$$

- The sulfate production rate was calculated using:
- $P[SO_A^{2-}] (\mu g m^{-3} h^{-1})$

$$= \frac{UVA \ (W \ m^{-2})}{850 \ (W \ m^{-2})} \times k_{obs} \ (mol \ L^{-1} \ s^{-1}) \times \frac{LWC \ (kg \ m^{-3})}{\rho \ (kg \ L^{-1})} \times 96 \ (g \ mol^{-1})$$

$$\times 3600 (s h^{-1}) \times 10^6 (\mu g g^{-1})$$

- where UVA is in W m<sup>-2</sup>,  $k_{obs}$  in mol L<sup>-1</sup> s<sup>-1</sup>, LWC in kg m<sup>-3</sup>, and  $\rho$  is the water density (kg L<sup>-1</sup>).
- Global distributions of SO<sub>2</sub>, LWC, and cloud pH were derived using the GEOS-Chem Classic v14.0.2 model
- (Bey et al., 2001), driven by MERRA-2 meteorological product from the Goddard Earth Observing System
- (GEOS) of the NASA Global Modeling and Assimilation Office (GMAO). SO<sub>2</sub> emissions were taken from
- the Community Emissions Data System (CEDS) inventory (Hoesly et al., 2018). Simulations were conducted
- at  $4^{\circ} \times 5^{\circ}$  resolution with 72 vertical layers from the surface to 0.01 hPa, and results were analyzed after a 2-
- 211 year spin-up.

### 3 RESULTS AND DISCUSSION

- 3.1 Photooxidation of Na<sub>2</sub>SO<sub>3</sub> solution under 370 nm irradiation. To simulate atmospheric aqueous-phase
- SO<sub>2</sub> oxidation, Na<sub>2</sub>SO<sub>3</sub> solutions with controlled initial pH were prepared and continuously bubbled with
- zero air (Fig. S1). At pH 4.0 and 0.5 mM Na<sub>2</sub>SO<sub>3</sub>, sulfite loss in the dark was slow, with an observed rate
- constant of 2.27 × 10<sup>-5</sup> s<sup>-1</sup> (Fig. S2) (Brandt and Eldik, 1995). Under UVA irradiation (370 nm; light spectrum

shown in Fig. S3A), the sulfite loss rate increased nearly tenfold to  $2.02 \times 10^{-4}$  s<sup>-1</sup>, with sulfate (SO<sub>4</sub><sup>2-</sup>) as the primary product (Fig. S4). Increasing Na<sub>2</sub>SO<sub>3</sub> concentration to 2.0 mM had only a moderate effect, with rate constants averaging ( $2.41 \pm 0.79$ ) ×10<sup>-4</sup> s<sup>-1</sup> (Fig. S5 and S6). In contrast, pH significantly influenced photooxidation kinetics (Fig. S7): the apparent sulfite decay rate increased by nearly 14-fold from pH 4.0 to 7.0, reflecting shifts in dominant S(IV) species (HSO<sub>3</sub><sup>-1</sup> vs. SO<sub>3</sub><sup>2-1</sup>) with different photochemical reactivities. These findings demonstrate that UVA light substantially enhances S(IV) oxidation in metal-free systems and that the reaction is strongly pH-dependent.

**3.2 Photodegradation of Guaiacol in Na<sub>2</sub>SO<sub>3</sub> solution.** Guaiacol (GUA), a methoxyphenol emitted primarily from biomass burning (4.7 Tg yr<sup>-1</sup> globally) (Liu et al., 2022; Li et al., 2023a), was used as a molecular probe to trace reactive intermediates formed during UVA-driven S(IV) oxidation. Given its Henry's law constant (Mcfall et al., 2020), up to 40% of atmospheric GUA can partition into the aqueous phase (Fig. S8), making it a relevant proxy for aqueous-phase organic transformations.

At pH 4.0, GUA (0.1 mM) was added to 2.0 mM  $Na_2SO_3$  solution under continuous zero-air bubbling. GUA remained stable in the dark, with minor losses attributed to evaporation (Fig. 1A). Under UVA irradiation (370 nm), however, it degraded rapidly following pseudo-first-order kinetics ( $k \approx 0.023 \, \text{min}^{-1}$ ; Fig. S9A) that is approximately 14 times faster than direct photolysis (Fig. S9B), highlighting the critical role of S(IV)-derived reactive intermediates. Suppressing  $O_2$  via  $N_2$  purging significantly reduced GUA degradation (Fig. 1A), and no degradation was observed under visible light (> 400 nm; Fig. S3B), confirming the importance of  $O_2$ -dependent photochemistry induced by UVA (Fig. S10).

We further investigated how reagent concentrations influence degradation kinetics. At high  $Na_2SO_3$ : GUA molar ratios ( $\geq 20$ ), GUA degradation followed pseudo-first-order kinetics, with rates increasing linearly with  $Na_2SO_3$  concentration (Fig. 1B). At lower ratios, deviations from first-order behavior were observed (Fig. S11), suggesting a shift in the limiting reagent or changes in radical propagation dynamics.

3.3 Formation of organosulfates and steady-state  $SO_4$  concentration. Figure S12 presents the kinetics of UVA-irradiated solutions containing 0.1 mM GUA and 0.5 mM  $Na_2SO_3$ . The apparent oxidation rate constant for  $SO_3^{2-}$  was  $6.34 \times 10^{-4}$  s<sup>-1</sup> (Fig. S13), about three times higher than that without GUA  $(2.02 \times 10^{-4} \text{ s}^{-1})$  (Fig. S4B), indicating that GUA significantly promoted S(IV) oxidation. The concurrent decrease in total inorganic sulfur closely tracked GUA degradation, suggesting that GUA reacted with photochemically generated intermediates to form S-containing organic species, such as organosulfates (OSs).

High-resolution mass spectrometry (HRMS;  $m/\Delta m = 5 \times 10^4$ ) was used to identify reaction products. Negative-mode ESI analysis of a solution containing 0.1 mM GUA and 2.0 mM Na<sub>2</sub>SO<sub>3</sub> at pH 4.0 (Fig. 2A) revealed unreacted GUA ( $C_7H_7O_2$ , m/z = 123.0446) along with multiple sulfate ester derivatives:  $C_6H_5O_5S^2$ (m/z = 188.9858),  $C_7H_7O_5S^2$  (m/z = 203.0014),  $C_7H_7O_6S^2$  (m/z = 218.9963), and  $C_7H_7O_7S^2$  (m/z = 234.9912).

These signals indicate OSs formation from GUA reacting with SO<sub>4</sub> radicals photochemically generated from 255 SO<sub>3</sub><sup>2-</sup> and O<sub>2</sub> under UVA. 256 To verify SO<sub>4</sub> involvement, we introduced 2,2,6,6-tetramethyl-1-piperinedinyloxy (TEMPO; C<sub>9</sub>H<sub>18</sub>NO) 257 as a radical scavenger (Bai et al., 2016). In the Na<sub>2</sub>SO<sub>3</sub> + TEMPO system without UVA (Fig. 2B) or after 30 258 minutes in the dark (Fig. 2C), only the TEMPO<sup>+</sup> -  $SO_3^{2-}$  adduct ( $C_9H_{18}NO_4S^-$ , m/z = 236.0968) was observed. 259 However, under 370 nm irradiation, new peaks appeared at m/z = 252.0903 and 220.1019, corresponding to 260 the TEMPO-SO<sub>4</sub> adduct (C<sub>9</sub>H<sub>18</sub>NO<sub>5</sub>S) and its O<sub>2</sub>-loss fragment (C<sub>9</sub>H<sub>18</sub>NO<sub>3</sub>S, Fig. 2D), respectively, 261 confirming SO<sub>4</sub> generation. 262 Finally, adding TEMPO to the GUA + Na<sub>2</sub>SO<sub>3</sub> + UVA system (Fig. 2E) eliminated all OS peaks, leaving 263 only signals for the TEMPO-SO<sub>4</sub> adduct and its fragment. This demonstrates that TEMPO scavenged SO<sub>4</sub> 264 and suppressed GUA-derived OS formation, confirming SO4. as the key intermediate driving the observed 265 OSs production. 266 SO<sub>4</sub> can also oxidize water or OH to form hydroxyl radicals (•OH) (Wojnárovits and Takács, 2019), which effectively oxidize GUA in aqueous phase (Yu et al., 2014). To assess the relative contributions of 267 268 •OH versus SO<sub>4</sub>·, we used ethanol (EtOH) and tert-butyl alcohol (tBuOH) as radical scavengers: EtOH reacts rapidly with both •OH (1.2 × 10<sup>9</sup> M<sup>-1</sup> s<sup>-1</sup>) and SO<sub>4</sub>• (1.6 × 10<sup>7</sup> M<sup>-1</sup> s<sup>-1</sup>), while tBuOH reacts primarily with 269 •OH  $(3.8 \times 10^8 \, \text{M}^{-1} \, \text{s}^{-1})$  and only weakly with  $SO_4$  ·  $(4 \times 10^5 \, \text{M}^{-1} \, \text{s}^{-1})$  (Liang and Su, 2009). At pH 4.0, adding 270 271 0.5 M EtOH significantly suppressed GUA photodegradation, while tBuOH had little effect, supporting SO<sub>4</sub>. 272 as the dominant oxidant (Fig. S14). 273 The second-order rate constant for GUA + SO<sub>4</sub> was determined using the relative rate method, with 274 phenol ( $k = 8.8 \times 10^9 \text{ M}^{-1} \text{ s}^{-1}$ ) as a reference (Fig. S15A) (Tran et al., 2022; Liang and Su, 2009). After accounting for direct photodegradation, the rate constant for GUA +  $SO_4$  was 3.78 ( $\pm$  0.42)  $\times$  10<sup>10</sup> M<sup>-1</sup> s<sup>-1</sup>, 275 276 which agrees well with the rate constant obtained from quantum chemical calculations (Li et al., 2023b). 277 From pseudo-first-order kinetics (Fig. 1B), the steady-state concentration of SO<sub>4</sub> under varying [Na<sub>2</sub>SO<sub>3</sub>] was estimated to be on the order of  $\sim 10^{-14}$  M (Fig. S15B). 278 279 280 3.4 Photochemical Pathway of SO<sub>4</sub>: formation from Na<sub>2</sub>SO<sub>3</sub> under UVA irradiation. In aqueous Na<sub>2</sub>SO<sub>3</sub>, the primary S(IV) species are SO<sub>2</sub>•H<sub>2</sub>O, HSO<sub>3</sub>, and SO<sub>3</sub><sup>2</sup>, with HSO<sub>3</sub> dominant under the experimental pH 281 282 range (Fig. S16). At pH 4.0, Na<sub>2</sub>SO<sub>3</sub> showed nearly no UV-vis absorption above 250 nm (Fig. 3A). Adding 283 0.1 mM GUA introduced a strong 274 nm peak from  $\pi - \pi^*$  transitions in GUA's aromatic ring. Although 284 initial UVA absorption was minimal, it increased markedly during irradiation, indicating the formation of 285 new light-absorbing products. 286 Since 370 nm UVA light (~ 3.35eV) lacks sufficient energy to directly excite either HSO<sub>3</sub> or the HSO<sub>3</sub>-287 GUA complex, the formation of SO<sub>4</sub>. likely involved photoactivation of intermediate complexes such as 288  $[HSO_3^- + O_2]$  or  $[SO_3^{2-} + O_2]$ . Time-dependent density functional theory (TDDFT) calculations support this,

showing that [SO<sub>3</sub><sup>2-</sup> + O<sub>2</sub>] can absorb UVA light (Fig. S17) and subsequently form reactive radicals. This is

consistent with previous findings that halide-O2 complexes can be photoexcited by UVA to yield radicals

291 like X• and HO<sub>2•</sub> (Cao et al., 2024a; Cao et al., 2024b).

- Density functional theory (DFT) calculations (Fig. 3B) reveal that electron transfer from the triplet state (T1) of  $[SO_3^2 + O_2]$  to form  $SO_3^-$  and  $O_2^+$  is endergonic (~ 13 kcal/mol) and unfavorable without light.
- TDDFT results indicate that UVA can excite T1 to higher-energy triplet states (T2), enabling this electron
- transfer. The resulting SO<sub>3</sub> is oxidized by O<sub>2</sub> to SO<sub>5</sub>, which decomposes to SO<sub>4</sub>, while O<sub>2</sub> further oxidizes
- S(IV) species.

298

307

310

313

290

292

293

#### 3.5 Mechanism of Guaiacol Photodegradation in Na<sub>2</sub>SO<sub>3</sub> Solutions Under UVA Irradiation

The photodegradation of GUA in aqueous Na<sub>2</sub>SO<sub>3</sub> solution under UVA irradiation proceeds through three major mechanisms, summarized in Table 1:

- 1) Formation of sulfur-centered radicals: Under UVA irradiation, the [SO<sub>3</sub><sup>2</sup> + O<sub>2</sub>] complex is photoexcited from the triplet state  $(T_1)$  to a higher triplet  $(T_2)$ , enabling electron transfer to produce  $SO_3$  and  $O_2^{\bullet}$ .  $SO_3^{\bullet}$  is rapidly oxidized by molecular  $O_2$  to form  $SO_5^{\bullet}$  at a high rate  $(k = 1.5 \times 10^9 \,\mathrm{M}^{-1} \,\mathrm{s}^{-1})$ .
- 2) Oxidation of sulfites to sulfate: SO<sub>5</sub> reacts with HSO<sub>3</sub> or SO<sub>3</sub><sup>2</sup> to produce SO<sub>4</sub>, SO<sub>3</sub> and SO<sub>5</sub><sup>2</sup>. SO<sub>3</sub> re-enters the cycle by reacting with O<sub>2</sub> to regenerate SO<sub>5</sub> SO<sub>5</sub> protonates to HSO<sub>5</sub>, which continues oxidizing S(IV) species to SO<sub>4</sub><sup>2</sup>. O<sub>2</sub> also oxidizes S(IV), but more slowly than SO<sub>5</sub> or SO<sub>4</sub> (Table S1). Importantly, SO<sub>5</sub>- reacts approximately 100 times faster with SO<sub>3</sub><sup>2</sup>- than with HSO<sub>3</sub>-, leading to a significant acceleration of sulfite photooxidation at pH > 4.0 where SO<sub>3</sub><sup>2-</sup> dominates (Fig. S7).
- 3) Formation of organosulfates:  $SO_4$  reacts with GUA extremely rapidly  $(k = 3.8 \times 10^{10} \text{ M}^{-1} \text{ s}^{-1})$ , much faster than with S(IV) species. This rapid reaction leads to substantial formation of low-volatility organics compounds, including OSs and GUA dimers and derivatives, with a SOA yield of ~ 80% (Fig. S18). GUA also increases the overall rate of sulfite oxidation by nearly threefold (Fig. S4B and S14), probably via additional reactive radicals generated during its reaction with SO<sub>4</sub>. Proposed mechanisms for the GUA-SO<sub>4</sub>. reaction are shown in Fig. S19 and S20.

315 316

319

- 3.6 Photodegradation of GUA at aqueous interfaces. In atmospheric environments, cloud and fog droplets typically range from a few to tens of micrometers in diameter. Within these microdroplets, surface-active solutes often concentrate at the air-water interface, where reactions are accelerated due to surface enrichment and reduced activation energies (Ruiz-Lopez et al., 2020). Classical molecular dynamics (MD) simulations revealed that GUA is energetically favored at the interface, with an interfacial free energy 2.8 kcal/mol lower than in bulk water (Fig. 4A and S21). SO<sub>4</sub> also shows an interfacial preference, although much smaller (0.17 kcal/mol difference) (Xie et al., 2024), suggesting that both species are enriched at the interface.
- Microdroplets also facilitate gas exchange, boosting [SO<sub>3</sub><sup>2-</sup> + O<sub>2</sub>] complex formation and SO<sub>4</sub> production 324 under UVA. Thus, GUA photodegradation is expected to be far greater in microdroplets than in bulk water - potentially by several orders of magnitude.

To test this, we used field-induced droplet ionization mass spectrometry (FIDI-MS) (Huang et al., 2018; Gong et al., 2022; Zhang et al., 2023) to monitor UVA-induced photodegradation of 0.1 mM GUA in microdroplets, with and without 3.0 mM Na<sub>2</sub>SO<sub>3</sub> (see Methods). Fig. 4B shows averaged FIDI-MS signals from five droplets, fitted to pseudo-first-order kinetics (Fig. S22 and S23). GUA degraded nearly 200 times faster at the interface than in bulk ( $k_{\text{bulk}} = 2.6 \times 10^{-5} \text{ s}^{-1} \text{ vs. } k_{\text{interface}} = 4.8 \times 10^{-3} \text{ s}^{-1}$ ). With Na<sub>2</sub>SO<sub>3</sub>, the rate similarly increased  $\sim 60$ -fold, indicating interfacial SO<sub>4</sub><sup>--</sup> concentrations of  $\sim 10^{-12}$  M, about two orders magnitude higher than in bulk.

Overall, these findings demonstrate that phenolic compounds like GUA are enriched and highly reactive at air-water interfaces, where UVA-driven SO<sub>4</sub>- formation greatly accelerates photodegradation and OS production.

### 3.7 Atmospheric Implications

- When gas-phase SO<sub>2</sub> dissolves into cloud and fog droplets, it hydrates to from S(IV) species such as SO<sub>3</sub><sup>2-</sup>. In the presence of O<sub>2</sub> and UVA irradiation, SO<sub>3</sub><sup>2-</sup> can be oxidized to SO<sub>4</sub><sup>2-</sup> through radical pathways. Using experimental data (Fig. S7), we derived a rate expression for sulfite oxidation at pH > 4.0. Under oxygen-saturated conditions ( $[O_2] = 2.5 \times 10^{-4} \text{ M}$ ), the rate constant is given by:
- Rate constant =  $1.75 \times 10^{-8} \times pH^{6.23}$  (s<sup>-1</sup>)
- Using these equations, we simulated UVA-driven sulfate formation under the AM0 standard solar spectrum and compared it with oxidation driven by conventional atmospheric oxidants (NO<sub>2</sub>, O<sub>3</sub>, and TMIs) (Cheng et al., 2016) (Fig. 5, see Methods). Under "Cloud droplets" conditions (John H. Seinfeld and Pandis, 2016; Herrmann et al., 2015) (Fig. 5A), UVA-driven sulfate formation in bulk solution (UVA<sub>low</sub>) is comparable in magnitude to conventional pathways, with its relative importance increasing in slightly acidic environments. Under "Beijing haze" conditions (Cheng et al., 2016), even with a 50% reduction in UVA intensity, UVA-induced sulfate formation can still dominate, particularly in the presence of small droplets (UVAhigh) (Fig. 5B).

We further modeled the global, seasonal  $SO_4^{2-}$  formation in atmospheric aqueous phases via S(IV) photooxidation of for the year 2023 (Fig. S24-28). At surface (Fig. 6) and at 1.29 km altitude (Fig. S29),  $SO_4^{2-}$  formation rates showed clear seasonal variation, with surface-level rates peaking at  $1.46 \times 10^{-2} \, \mu g \, m^{-3} \, h^{-1}$ . Higher rates were observed in regions such as Asia and South Africa, driven by elevated  $SO_2$  emissions from industry and power generation and strong solar radiation during warmer months. In North America and the Mediterranean, despite lower  $SO_2$  levels, higher aqueous-phase pH promotes S(IV) accumulation and accelerates oxidation rates. Over marine regions, limited direct  $SO_2$  emissions lead to lower S(IV) concentrations than over the land. However, at 1.29 km, the greater availability of liquid water makes these areas stable and significant zones for sulfate formation, with production rates ( $P[SO_4^{2-}]$ ) ranging from  $10^{-6}$  to  $10^{-4} \, \mu g \, m^{-3} \, h^{-1}$ .

368

374375

376377

380

384

386

388

390

394

In summary, our results reveal a previously unrecognized, metal-free pathway for SO2 oxidation to sulfate in atmospheric aqueous phases under UVA irradiation. Unlike traditional mechanisms that rely on metal catalysts or high-energy UVB/UVC lights, we show that the [SO<sub>3</sub><sup>2-</sup> + O<sub>2</sub>] complex can initiate sulfate radical production under UVA – wavelengths far more prevalent in the solar spectrum. In the presence of guaiacol - a common phenolic compound from biomass burning, these sulfate radicals drive rapid GUA oxidation, producing low-volatility organic compounds, including organosulfates, at a high second-order rate constant of  $3.8 \times 10^{10}~M^{\text{--}1}~\text{s}^{\text{--}1}$ . Moreover, microdroplet experiments show that GUA photodegradation is dramatically accelerated in small droplets under UVA light due to intensified interfacial chemistry. The high surface-area-to-volume ratio of microdroplets promotes efficient generation of reactive oxidants, particularly sulfate radicals, which accelerate both S(IV) oxidation and organics transformations. Together, these findings uncover a novel, sunlight-accessible, metal-free pathway for sulfate and SOA formation, especially relevant to slightly acidic, sunlit, and water-rich atmospheric environments. **Supplement.** The supplement related to this article is available online. Data availability. The data that support the findings of this study are available in the supplement of this article. Author contributions. B.C., X.Y., and Q.Z. designed research; B.C., Y.H., W.J., X.Y., and Q.Z. performed research; B.C., Y.H., W.J., Y.L., Y.L., J.Z., Y.Z., J.Y., H.S., C.W., L.Z., T.F., Q.Z., and X.Y. analyzed data; B.C., Y.H., Q.Z., and X.Y. wrote the paper. Competing interests. The contact author has declared that none of the authors has any competing interests. Acknowledgments. Supported by Center for Computational Science and Engineering at Southern University of Science and Technology. Qi Zhang acknowledges support from the Donald G. Crosby Endowed Chair at the University of California at Davis. Financial support. This work was supported by Shenzhen Key Laboratory of Precision Measurement and Early Warning Technology for Urban Environmental Health Risks (ZDSYS20220606100604008), Guangdong Provincial Observation and Research Station for Coastal Atmosphere and Climate of the Greater Bay Area (2021B1212050024), Shenzhen Science and Technology Program (KQTD20210811090048025, KCXFZ20230731093601003).

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

- Fast Hydroxyl Radical Generation at the Air-Water Interface of Aerosols Mediated by Water-

# https://doi.org/10.5194/egusphere-2025-5323 Preprint. Discussion started: 12 November 2025 © Author(s) 2025. CC BY 4.0 License.

- Soluble PM<sub>2.5</sub> under Ultraviolet A Radiation, J. Am. Chem. Soc., 145, 6462–6470,
- 10.1021/jacs.3c00300, 2023.
- Zhang, R. and Chan, C. K.: Simultaneous formation of sulfate and nitrate via co-uptake of SO<sub>2</sub>
- and NO<sub>2</sub> by aqueous NaCl droplets: combined effect of nitrate photolysis and chlorine
- chemistry, Atmos. Chem. Phys., 23, 6113–6126, 10.5194/acp-23-6113-2023, 2023.
- Zhao, J., Zheng, B., Ciais, P., Chen, Y., Gasser, T., Canadell, J. G., Zhang, L., and Zhang, Q.: Global
- warming amplifies wildfire health burden and reshapes inequality, Nature, 10.1038/s41586-
- 41025-09612-41589, 10.1038/s41586-025-09612-9, 2025.
- Zhao, Y. and Truhlar, D. G.: The M06 suite of density functionals for main group
- thermochemistry, thermochemical kinetics, noncovalent interactions, excited states, and
- transition elements: two new functionals and systematic testing of four M06-class functionals
- and 12 other functionals, Theor. Chem. Acc., 120, 215–241, 10.1007/s00214-007-0310-x, 2007.
- Zheng, B., Zhang, Q., Zhang, Y., He, K. B., Wang, K., Zheng, G. J., Duan, F. K., Ma, Y. L., and
- Kimoto, T.: Heterogeneous chemistry: a mechanism missing in current models to explain
- secondary inorganic aerosol formation during the January 2013 haze episode in North China,
- Atmos. Chem. Phys., 15, 2031–2049, 10.5194/acp-15-2031-2015, 2015.
- Zheng, J., Xu, X., and Truhlar, D. G.: Minimally augmented Karlsruhe basis sets, Theor. Chem.
- Acc., 128, 295–305, 10.1007/s00214-010-0846-z, 2010.
- Zuo, Y., Zhan, J., and Wu, T.: Effects of Monochromatic UV-Visible Light and Sunlight on Fe(III)-
- Catalyzed Oxidation of Dissolved Sulfur Dioxide, J. Atmos. Chem., 50, 195-210, 10.1007/s10874-
- 005-2813-y, 2005.

# 644 Figures and Tables

## 645 Figure 1.

(A) Kinetics of the aqueous-phase reaction between guaiacol and Na<sub>2</sub>SO<sub>3</sub> under different conditions. (B) The dependence of the pseudo-first-order rate constant for GUA decay on the concentration of Na<sub>2</sub>SO<sub>3</sub>. Error bars represent the standard deviations from independent experiments. Experimental conditions: [guaiacol] = 0.1 mM, [Na<sub>2</sub>SO<sub>3</sub>] = 2.0 mM, pH =  $4.0 \pm 0.1$ , zero-air bubbling, 370 nm light irradiation, room temperature.

651

647

648

# 652 Figure 2.

High-resolution mass spectra of reaction products from: (A) GUA +  $Na_2SO_3$  under after 30 min of 370 nm irradiation; (B)  $Na_2SO_3$  + TEMPO at 0 min; (C)  $Na_2SO_3$  + TEMPO after 30 min in the dark; (D)  $Na_2SO_3$  + TEMPO after 30 min of 370 nm irradiation; and (E) GUA +  $Na_2SO_3$  + TEMPO after 30 min of 370 nm irradiation. Experimental conditions: [guaiacol] = 0.1 mM, [ $Na_2SO_3$ ] = 2.0 mM, [TEMPO] = 4.0 mM, pH =  $4.0 \pm 0.1$ , zero-air bubbling, and room temperature. Proposed chemical structures corresponding to the key mass spectral peaks are shown to the right of the spectra.

# 661 Figure 3.

662663

664

665

666

667

(A) UV-vis absorption spectra of the GUA +  $Na_2SO_3$  reaction solution at different time points. (B) Gibbs free energy profiles (kcal/mol, 298.15 K) for the  $SO_3^{2-}$  +  $O_2$  reaction, calculated at the CCSD(T)/aug-cc-pVTZ/SMD(water)//M06-2X/ma-TZVP/SMD(water) level with Zero Point Energy (ZPE) correction, with the inset showing the vertical excitation spectra of the  $[SO_3^{2-} + O_2]$  complex, calculated using TDDFT at the M06-2X/ma-TZVP/SMD(water) level.

# 669 Figure 4.

671

672

673

674

(A) Free energy profiles for GUA transfer from the gas phase to bulk water, overlaid with water density distribution at air-water interface. (B) Kinetics of direct photodegradation of GUA in microdroplets, with and without Na<sub>2</sub>SO<sub>3</sub>, under UVA irradiation.

# 675 Figure 5.

Simulated aqueous-phase sulfate production rates from SO<sub>2</sub> oxidation as a function of pH under two atmospheric scenarios: (A) "Cloud droplets" scenario with full UVA intensity (AM0 standard). (B) "Beijing haze" scenario with 50% reduced UVA intensity. Colored lines represent contributions from individual oxidants and the shaded region indicates the total sulfate production range bounded by the UVA high and

UVA<sub>low</sub> estimates.

677

678

679

680

681

# 683 Figure 6.

 $P[SO_4{}^{2-}]\;\; in \; aqueous-phase at surface from GEOS-Chem <math display="inline">[\mu g/m^3/h]$ 

Seasonal variations in surface-level aqueous-phase SO<sub>4</sub><sup>2-</sup> formation rates from S(IV) photoreactions in 2023.

685 686

Table 1. Reactions and rate constants of GUA photodegradation in Na<sub>2</sub>SO<sub>3</sub> solutions (John H. Seinfeld and
 Pandis, 2016; Rudzinski et al., 2009).

| 1) Formation of sulfur radical                                                |                                                 |
|-------------------------------------------------------------------------------|-------------------------------------------------|
| $HSO_3^- \rightarrow SO_3^{2-} + H^+$                                         | $6.75 \times 10^3 \text{ s}^{-1}$               |
| $SO_3^{2-} + H^+ \rightarrow HSO_3^-$                                         | $1.0 \times 10^{11}  M^{-1}  s^{-1}$            |
| $SO_3^{2-} + O_2 \rightarrow [SO_3^{2-} + O_2]$                               |                                                 |
| $[SO_3^{2-} + O_2] + UVA \rightarrow SO_3^{-} + O_2^{-}$                      |                                                 |
| $SO_3^{\bullet} + O_2 \rightarrow SO_5^{\bullet}$                             | $1.5 \times 10^9  M^{1}   \mathrm{s}^{1}$       |
| 2) Oxidation of sulfites                                                      |                                                 |
| $SO_5^{\bullet -} + HSO_3^{-} \rightarrow SO_3^{\bullet -} + SO_5^{2-} + H^+$ | $2.5 \times 10^4  M^{1}  \text{s}^{1}$          |
| $SO_5^{\bullet -} + HSO_3^{-} \rightarrow SO_4^{\bullet -} + SO_4^{2-} + H^+$ | $7.5 \times 10^4  M^{1}   \text{s}^{1}$         |
| $SO_5^{\bullet -} + SO_3^{2-} \longrightarrow SO_3^{\bullet -} + SO_5^{2-}$   | $3.25 \times 10^6 \ M^{1} \ \text{s}^{1}$       |
| $SO_5^{\bullet -} + SO_3^{2-} \longrightarrow SO_4^{\bullet -} + SO_4^{2-}$   | $9.75 \times 10^6  M^{1}  \text{s}^{1}$         |
| $SO_4$ + $HSO_3$ $\rightarrow SO_3$ + $SO_4$ + $H$                            | $1.4 \times 10^7  \text{M}^{1}  \text{s}^{1}$   |
| $SO_4$ + $SO_3$ <sup>2-</sup> $\rightarrow SO_3$ + $SO_4$ <sup>2-</sup>       | $1.4 \times 10^7  M^{1}  \text{s}^{1}$          |
| $SO_5^{2-} + H^+ \rightarrow HSO_5^-$                                         | $1.0 \times 10^{10}  M^{1}  s^{1}$              |
| $\mathrm{HSO_5^-} \rightarrow \mathrm{SO_5^{2-}} + \mathrm{H^+}$              | 3.98 s <sup>-1</sup>                            |
| $HSO_5^- + HSO_3^- \rightarrow 2SO_4^{2-} + 2H^+$                             | $7.5 \times 10^3 \text{ M}^{-1} \text{ s}^{-1}$ |
| $HSO_5^- + SO_3^{2-} \rightarrow 2SO_4^{2-} + H^+$                            | $7.5 \times 10^3 \text{ M}^{-1} \text{ s}^{-1}$ |
| 3) Formation of OSs                                                           |                                                 |
| GUA + SO₄*- → Products (including OSs)                                        | $3.8 \times 10^{10} \ M^{1} \ s^{1}$            |