# Peer review of "Rapid Secondary Organic Aerosol Formation at the Air–Water Interface from"

_EGUsphere, 2025_

## Author Comment (AC1)

We appreciate the reviewers' valuable comments and have revised the manuscript accordingly. Listed below are our point-to-point responses (in blue) to each comment (in black) and corresponding changes in the manuscript indicated in red.

**Reviewer #1:**

*Comments for the Author:*

*This study combines laboratory experiments with quantum chemical calculations to investigate a novel, metal-free UVA-induced pathway for $SO_4^{\bullet-}$ generation. Using guaiacol (GUA), a representative biomass-burning phenol, as a molecular probe, the authors tracked radical activity and organosulfate (OS) formation. Notably, microdroplet experiments reveal a 100-fold rate enhancement, attributed to interfacial effects. Modeling further suggests that this aqueous UVA pathway represents a significant source of sulfate in the atmosphere. The manuscript is well written, and the results are both new and exciting. I support publication, subject to the following minor comments and suggestions:*

We appreciate the positive comments from the review.

*Specific Comments*
*Line 122:*

*Please define "FIDI-MS." Additionally, for the FIDI-MS experiments, what was the ambient relative humidity (RH) in the laboratory? What were the composition and concentrations of species in the suspended droplets? Would they be different from that of bulk stock solutions?*

FIDI-MS refers to field-induced droplet ionization mass spectrometry. In the revised manuscript, we have added a definition of FIDI-MS and cited the relevant reference. A brief description of the technique is now included in the Methods section to clarify its working principle.

The ambient relative humidity (RH) during the FIDI-MS experiments was approximately 50%, which is representative of typical laboratory conditions.

Regarding the suspended droplets, their chemical composition and solute concentrations were the same as those of the corresponding bulk stock solutions used for droplet generation. No additional species were introduced during the droplet formation or ionization process. Therefore, the composition and concentrations of species in the suspended droplets are not expected to differ from those in the bulk solutions. We have clarified this point in the revised manuscript to avoid potential ambiguity.

The texts read:
FIDI-MS experiment. The working principle of FIDI-MS (field-induced droplet ionization mass spectrometry) is described as follows (Gong et al., 2022). Droplets approximately 2 mm in diameter (~4 μL volume) were suspended from the tip of a stainless-steel capillary, which was positioned equidistantly between two parallel plate electrodes separated by 6.3 mm apart. Droplets were formed by injecting the analyte solution through the capillary using a syringe pump. The chemical composition and solute concentrations of the suspended droplets were identical to those of the corresponding bulk stock solutions used for droplet generation. The parallel plates were mounted on a translation stage to align the front electrode's aperture with the atmospheric pressure inlet of a Thermo-Fischer LTQ-XL mass spectrometer (Waltham,

MA), which was operated under laboratory ambient air conditions at a relative humidity of approximately 50%. Once the droplets were formed, a 60-second equilibration period was allowed to enable compound diffusion and achieve equilibrium coverage at the air-water interface.

*Line 122:*

"*Droplets approximately 2 mm in diameter (~4 µL volume) were suspended from the tip of a stainless-steel capillary, positioned equidistantly between two parallel plate electrodes separated by 6.3 mm.* " *Does droplet size affect reaction rates? Furthermore, how can these findings be extrapolated to submicron-sized droplets?*

In the present study, we did not systematically investigate the effect of droplet size on reaction rates. The droplet size used in the FIDI-MS experiments (~2 mm in diameter) was chosen based on the technical requirements of stable droplet suspension and signal reproducibility, and was kept constant throughout the experiments.

It should be noted that FIDI-MS is primarily designed to probe chemical processes occurring at the air–water interface, rather than to quantitatively reproduce reaction kinetics in submicron-sized droplets. The key objective of using FIDI-MS in this work is to characterize interfacial reaction pathways and product formation under controlled conditions, which are expected to be qualitatively relevant to atmospheric aqueous interfaces.

We acknowledge that submicron droplets may exhibit enhanced surface-to-volume ratios and potentially different kinetic behaviors, and direct extrapolation of absolute reaction rates from millimeter-sized droplets to submicron aerosols should therefore be made with caution. Investigation of size-dependent effects and reactions in submicron droplets would require dedicated experimental platforms, such as aerosol generators and flow-tube or chamber-based measurements. We consider this an important direction for future work and plan to explore it using complementary experimental approaches.

*Line 152:*

*For Section 2.3 (theoretical calculations), were droplet size or curvature effectsconsidered in the DFT, TDDFT, and MD calculations?*

In Section 2.3, the droplet size or curvature effects were not explicitly considered in the DFT, TDDFT, or MD calculations. Because our current experiments were not conducted on submicron-sized droplets, we chose not to introduce additional model complexity related to droplet curvature or size-dependent effects. This approach allows us to focus on elucidating the intrinsic reaction pathways and electronic properties under well-defined bulk aqueous conditions, which serve as a baseline for understanding the observed air–water interfacial chemistry.

We acknowledge that submicron droplets may experience enhanced interfacial electric fields (Hao et al., 2022), which could influence molecular orientation, reaction energetics, and kinetics. Such electric field effects were not included in the present DFT, TDDFT, or MD simulations. Quantitative treatment of curvature- and field-induced effects would require specialized interfacial models and dedicated experimental

validation, which is beyond the scope of the current study but represents an important direction for future work.

*Line 179:*

*For the box model conditions, since experimental measurements were conducted with 2 mm droplets, should potential size effects on reaction rates be considered if they are significant? Additionally, should other factors─such as concentration variations and the presence of inorganic and organic species in atmospheric droplets ─ be incorporated to better represent atmospheric conditions in the simulations?*

The box model simulations in this study were intended to isolate and evaluate the relative importance of UVA-driven S(IV) oxidation, rather than to explicitly reproduce the full chemical complexity of atmospheric cloud or fog droplets.

To better constrain the bulk-phase reaction rates used in the simulations, we recalculated the apparent photon efficiency (APE) based on our experimental UVA results. This approach allows a more physically consistent parameterization of the bulk aqueous oxidation kinetics under UVA irradiation, thereby improving the representation of photochemical S(IV) oxidation in the box model framework.

We acknowledge that droplet size effects, particularly potential rate enhancements in micron- or millimeter-scale cloud and fog droplets, as well as the presence of coexisting inorganic and organic constituents, can substantially influence reaction kinetics in real atmospheric systems. However, robust incorporation of these effects into the model requires experimentally constrained rate constants that explicitly account for droplet size, interfacial processes, and multiphase chemical interactions. At present, such parameters are not sufficiently constrained. We therefore plan to conduct additional experiments under atmospherically relevant droplet sizes and compositional conditions to derive more representative rate constants, which will enable more realistic treatment of these processes in future modeling efforts.

*Line 232:*

*"At pH 4.0, GUA (0.1 mM) was added to 2.0 mM $Na_2SO_3$ solution under continuous zero-air bubbling." Can the authors justify the chosen GUA concentration? Is it atmospherically relevant?*

We thank the reviewer for this question regarding the atmospheric relevance of the chosen guaiacol (GUA) concentration. The concentration of GUA used in this study (100 μM) represents an upper-bound but plausible aqueous-phase concentration under biomass burning–influenced polluted conditions.

Based on GUA's Henry's law constant ($4.8 \times 10^3$ M atm$^{-1}$ at 5 °C (Mcfall et al., 2020)), approximately 3% of gas-phase GUA is expected to partition into cloud or fog water with a liquid water content of 0.3 g $H_2O$ m$^{-3}$ under Henry's law equilibrium (Sun et al., 2010). However, field observations indicate substantially enhanced aqueous-phase partitioning. For example, measurements conducted in urban California during biomass burning events show that 9–11% of total phenols was present in fog droplets, corresponding to enhancement factors of approximately 3–4 relative to equilibrium solubility estimates (Sagebiel and Seiber, 1993).

In addition, a GUA concentration of 0.1 mM was selected for this study to ensure sufficient analytical sensitivity for mass spectrometric analysis and to maintain consistency with prior laboratory studies of atmospheric aqueous-phase reactions of phenolic compounds (Yu et al., 2014; Smith et al., 2015; Pang et al., 2019a; Pang et al., 2019b; Ma et al., 2021; Jiang et al., 2021; Arciva et al., 2022; Mabato et al., 2023). This concentration represents a realistic upper-limit scenario for phenolic compounds in atmospheric aqueous microenvironments, while allowing for robust mechanistic characterization of aqueous-phase reactions.

*Line 239:*

"*We further investigated how reagent concentrations influence degradation kinetics. At high $Na_2SO_3$:GUA molar ratios ($\geq 20$), ...*" *What would typical atmospheric ratios be?*
Reported aqueous-phase concentrations of GUA in previous studies can reach up to $\sim 10^{-4}$ M, particularly in regions influenced by biomass burning, whereas in non-fire-impacted environments, GUA concentrations are generally lower.

Atmospheric $SO_2$ mixing ratios vary widely, ranging from a few ppb in background regions to several tens of ppb in polluted or biomass-burning-influenced areas. Based on Henry's law equilibrium, these gas-phase $SO_2$ levels correspond to estimated aqueous-phase S(IV) concentrations spanning approximately $10^{-6}$ to $10^{-3}$ M, depending on $SO_2$ abundance and aqueous conditions. Consequently, the atmospheric S(IV) : GUA molar ratio can vary by several orders of magnitude across different environments.

To account for this variability, we performed experiments over a broad range of $Na_2SO_3$ : GUA molar ratios, including high ratios ($\geq 20$), as shown in Fig. S14. This experimental design allows us to assess how relative reactant abundances influence GUA degradation kinetics under different, but atmospherically plausible, conditions.

*Line 254:*

"*These signals indicate OS formation from GUA reacting with $SO_4{}^{\bullet-}$ radicals photochemically generated from $SO_3{}^{2-}$ and $O_2$ under UVA.*" *What are their formation mechanisms for the detected OS species?*
The detected OS species are formed via the reaction of guaiacol (GUA) with sulfate radicals ($SO_4{}^{\cdot-}$). The possible formation mechanisms of the OS species, as illustrated in Fig. S22 and Fig. S23 of the Supporting Information, involve direct radical–radical coupling and addition of sulfate radicals to the aromatic ring of guaiacol. These pathways are consistent with the observed product distributions and support the assignment of the detected OS signals.

*Line 256:*

"*To verify $SO_4{}^{\bullet-}$ involvement, we introduced 2,2,6,6-tetramethyl-1-piperidinyloxy (TEMPO; $C_9H_{18}NO$) as a radical scavenger (Bai et al., 2016).*" *Would the presence of OH radicals in the aqueous phase affect the results?*

To verify the involvement of sulfate radicals (SO$_4$$^{·-}$), TEMPO (2,2,6,6-tetramethyl-1-piperidinyloxy) was introduced as a selective radical scavenger. Upon addition of TEMPO, we observed corresponding mass spectral peaks (C$_9$H$_{18}$NO$_3$S$^-$ and C$_9$H$_{18}$NO$_5$S$^-$) in the HRMS measurements (Figure 2), providing direct evidence for the presence and reactivity of SO$_4$$^{·-}$ in the reaction system.

To evaluate potential interference from hydroxyl radicals (·OH), we also conducted experiments using methanol and tert-butanol, which exhibit different scavenging efficiencies towards SO$_4$$^{·-}$ and ·OH (Fig. S17). The negligible changes in product formation upon ·OH quenching indicate that hydroxyl radicals play a negligible role under our experimental conditions. These results demonstrate that SO$_4$$^{·-}$ is the dominant reactive species in the system and that the presence of aqueous-phase ·OH does not significantly affect the reaction outcomes.

*Line 280:*

*"Photochemical Pathway of SO$_4$•$^-$ formation from Na$_2$SO$_3$ under UVA irradiation."*
*Were droplet size and curvature effect considered?*
The photochemical pathway of SO$_4$$^{·-}$ formation from Na$_2$SO$_3$ under UVA irradiation was investigated using UV–vis absorption measurements combined with TDDFT calculations to elucidate the fundamental mechanism of sulfate radical generation.

In this part of the study, droplet size and curvature effects at the submicron scale were not considered. The experiments and calculations focused on bulk aqueous solutions and macroscopic air – water interfaces, which provide the mechanistic basis for understanding interfacial sulfate radical formation. Extension of this work to submicron-sized droplets—where size- and curvature-dependent effects may become significant—will be addressed in future studies.

*Line 323:*

*"Microdroplets also facilitate gas exchange, boosting [SO$_3$$^{2-}$ + O$_2$] complex formation and SO$_4$•$^-$ production under UVA. Thus, GUA photodegradation is expected to be far greater in microdroplets than in bulk water—potentially by several orders of magnitude."* *Is the enhancement primarily attributed to gas exchange due to change in droplet size?*
Based on our experiments and TDDFT analysis, the enhanced GUA photodegradation observed in microdroplets is primarily driven by the UVA-induced formation of [SO$_3$$^{2-}$ + O$_2$] complexes, which subsequently generate sulfate radicals (SO$_4$$^{·-}$).

At the air–water interface, GUA exhibits a lower reaction energy (-2.8 kcal/mol) than in bulk water, making the reaction thermodynamically more favorable (Figure 4A). Additionally, the interface promotes efficient gas–liquid O$_2$ exchange, leading to higher local concentrations of [SO$_3$$^{2-}$ + O$_2$] complexes and consequently greater SO$_4$$^{·-}$ production. Therefore, the observed enhancement arises from the combined effects of favorable interfacial energetics of GUA and increased O$_2$ availability.

*Line 326:*

*"To test this, we used field-induced droplet ionization mass spectrometry (FIDI-MS) (Huang et al., 2018; Gong et al., 2022; Zhang et al., 2023) to monitor UVA-induced photodegradation of 0.1 mM GUA in microdroplets, with and without 3.0 mM $Na_2SO_3$ (see Methods)."* Were the compositions identical in both cases?

Yes, the droplet compositions were identical in both experiments, with the sole difference being the presence or absence of 3.0 mM $Na_2SO_3$. Experiments without $Na_2SO_3$ were designed to probe the direct photolysis of GUA under UVA, while experiments with $Na_2SO_3$ were conducted to investigate the enhancement of GUA photodegradation under the same UVA irradiation conditions. All other experimental parameters, including droplet size, ambient conditions, and GUA concentration (0.1 mM), were kept constant.

**Reviewer #2:**

*The manuscript investigates a proposed metal-free photochemical mechanism for generating sulfate radical anions (SO4-.) via the UVA irradiation of aqueous S(IV)-oxygen complexes. However, the work suffers from significant technical and conceptual deficiencies that undermine its primary conclusions, most notably the absence of measured photochemical quantum efficiency. Without this fundamental parameter, the use of an artificial light source with an intensity over ten times that of natural sunlight (850 W/m2) makes the atmospheric scaling and claims of global significance speculative and physically unbenchmarked. Furthermore, the assertion of a "novel" sunlight-driven pathway is contradicted by established literature (e.g., Galloway et al., 2009; Nozière et al., 2010; Cope et al., 2022) that has already documented radical-mediated sulfate chemistry under similar conditions. The reported second-order rate constants (3.78\*10^10 1/M 1/s) are also at or beyond the physical diffusion limit, suggesting potential experimental artifacts or the influence of trace impurities. Additionally, the modeling fails to account for competing established pathways, such as transition metal ion (TMI) catalysis and light attenuation by black carbon, which likely results in an overestimation of this pathway's dominance. The recommendation to the editor is to reject the manuscript in its current form. The manuscript requires a fundamental re-evaluation of its novelty, the inclusion of rigorous actinometry to determine quantum efficiency, and a balanced comparative analysis against other works before any potential reconsideration.*

We respectfully disagree with the reviewer's overall assessment and recommendation for rejection. We nevertheless appreciate the detailed and critical evaluation of our manuscript, and below we clarify several key points regarding our experimental design, the treatment of light intensity and quantum efficiency, and the scope and intent of the modeling analysis.

First, we acknowledge that a formal photochemical quantum efficiency for the $[SO_3^{2-} + O_2]$ complex cannot be rigorously determined with current methodologies. This limitation is inherent to this class of relative, transient complexes rather than a shortcoming of experimental design. The $[SO_3^{2-} + O_2]$ complex is transient, and its steady-state concentration cannot be independently quantified. Unlike classical photochemical systems such as $H_2O_2$ or HONO—where the parent molecules are stable and photolysis proceeds as a largely isolated one-step process— the photolysis of $[SO_3^{2-} + O_2]$ initiates a coupled catalytic radical chain mechanism. The initially generated $SO_3^{\cdot-}$ rapidly reacts with $O_2$ to form $SO_5^{\cdot-}$, which efficiently oxidizes $SO_3^{2-}$ and propagates further radical chemistry (Figure 1). As a result, photolysis products actively consume the reactant, making it impossible to isolate a single elementary photolysis step required for rigorous determination of quantum yield.

[Figure]

Figure 1. The mechanism of S(IV) oxidation triggered by free radicals

The photolysis of the $[SO_3^{2-} + O_2]$ complex therefore represents only the initiating elementary step of a coupled radical chain process. In principle, one could attempt to deconvolute the apparent kinetics into a full set of elementary reactions and back-calculate the initiating photolysis rate and quantum yield. However, this approach is not currently viable because reported rate constants for key S(IV) radical reactions span 2–4 orders of magnitude in the literature, including authoritative atmospheric chemistry references (John H. Seinfeld and Pandis, 2016; Das, 2001). Without a consistent and well-constrained kinetic framework, such back-calculation would introduce substantially larger uncertainties than those associated with the apparent-rate framework adopted here.

Recognizing this constraint, we instead quantified an apparent photon efficiency (APE) based on experimentally measured sulfate formation rates under controlled UVA irradiation. This APE provides an experimentally anchored metric that integrates photon absorption, photochemical initiation efficiency, and subsequent radical chain propagation. For multiphase radical systems in which photolysis serves as an initiating rather than terminal step, such apparent efficiencies have been widely used to assess photochemical relevance. Importantly, the APE was subsequently used to rescale reaction rates in the box model by explicitly accounting for differences in photon flux between the laboratory UVA source and natural solar irradiation, rather than assuming direct equivalence.

With respect to light intensity, elevated laboratory irradiance was employed to enable observation of low-probability photochemical initiation processes within experimentally accessible timescales. Reaction rates were normalized by photon exposure, and the modeling explicitly accounts for the lower photon flux under natural solar conditions. We emphasize that no claim is made that laboratory irradiance directly mimics ambient sunlight, but rather that photon-normalized kinetics provide a physically meaningful basis for atmospheric scaling.

We further clarify that the objective of this study is not to claim dominance of this pathway over all established sulfate formation mechanisms, nor to provide a quantitative global sulfate budget. Rather, our work identifies and mechanistically constrains a previously underexplored UVA-driven S(IV) oxidation pathway that is directly coupled to phenolic compound photodegradation. While radical-mediated sulfate chemistry has been reported previously, prior studies have primarily focused on metal-catalyzed, or non-UVA-driven processes. The specific coupling between UVA excitation, $[SO_3^{2-} + O_2]$ complex photochemistry, and phenolic compound degradation demonstrated here has not been explicitly demonstrated in earlier work.

The reported second-order rate constants reflect effective rates within a coupled radical chain system rather than isolated bimolecular diffusion-limited reactions. As such, their magnitude should be interpreted in the context of chain propagation rather than diffusion-limited collision rates. Potential contributions from trace metal impurities were minimized through rigorous reagent selection and control experiments, and the potential influence of transition metal ion catalysis and light attenuation by black carbon are discussed in the revised manuscript as important complementary processes rather than neglected pathways.

Taken together, we respectfully submit that the concerns raised reflect the inherent challenges associated with studying multiphase photochemical radical systems rather than grounds for rejection. Although classical quantum yield determination is not feasible for this system, the combined use of controlled laboratory experiments, TDDFT analysis, and photon-normalized modeling provides internally consistent and physically constrained mechanistic insight. The revised manuscript clarifies its scope, assumptions, and limitations, and places the proposed pathway in appropriate context relative to established sulfate formation mechanisms. We therefore respectfully suggest that the revised manuscript merits reconsideration.

*Major Comments:*

*1) The abstract claims to identify a "novel, metal-free mechanism" for generation of reactive species for sulfate formation under UVA. However, existing literature (e.g., Gong et al., 2022 and others) already suggests that UVA light promotes oxidation at the interface. The claim of absolute novelty conflicts with studies exploring S(IV) photo-excitation.*

*Cope et al. (2022) provided the first direct experimental evidence that UV irradiation of aqueous sulfate, whether natural sunlight or lab-based UV, generates sulfate radical anion, enabling oxidation of organic compounds even under typical tropospheric pH and ionic strength conditions.*

*Nozi`ere et al. (2010) were among the first to report UV-254 nm (UV-B) irradiation of ammonium sulfate with alkenes, yielding organosulfates via radical-mediated pathways. Although the precise mechanism wasn't specified, these results imply sulfate radical involvement.*

*Galloway et al. (2009) observed the light-triggered production of organosulfates, such as glycolic acid sulfate, during glyoxal uptake onto ammonium sulfate aerosols under UVA, but not in the dark, supporting the presence of photogenerated sulfate-radical*

*chemistry.*

We would like to clarify that our study focuses on a metal-free, UVA-induced oxidation of aqueous S(IV) ($HSO_3^-/SO_3^{2-}$ at pH 4-7) that generates sulfate radicals ($SO_4^{\cdot-}$), which subsequently react with guaiacol (GUA) to form organosulfates. This process is distinct from previously reported photochemistry of high-concentration sulfate solutions (e.g., Cope et al., 2022, Nozière et al., 2010, Galloway et al., 2009), where $SO_4^{\cdot-}$ is generated directly from UV-irradiated $SO_4^{2-}$ (3.7 M). In contrast to these studies, our work does not involve direct photolysis of sulfate at molar concentrations. Instead, it addresses a parallel and distinct pathway in while sulfate radicals are produced via UVA-induced oxidation of S(IV) at environmentally relevant pH (4-7) and much lower sulfur concentrations. This distinction is important because S(IV) species ($HSO_3^-/SO_3^{2-}$) are common in atmospheric aqueous phases and exhibit different photochemical behavior than $SO_4^{2-}$.

Compared to Gong et al. (2022), who reported UVA-enhanced interfacial $SO_2$ oxidation, our system also differs in both chemical regime and mechanism. We intentionally restrict our experiments to pH 4-7, as bubbling zero air through $Na_2SO_3$ solutions at pH < 4 leads to substantial sulfur loss due to volatilization of $SO_2$. Moreover, while Gong et al. emphasized $\cdot OH$-driven chemistry, we show that quenching $\cdot OH$ with excess tBuOH does not suppress GUA degradation, indicating that $SO_4^{\cdot-}$ formation and reactivity in our system proceed independently of $\cdot OH$.

Thus, the mechanism in our work is fundamentally different from previously reported UVA-driven pathways: it is a metal-free, UVA-induced S(IV) oxidation at environmentally relevant pH, and its specific coupling to GUA degradation and organosulfate formation. This mechanism complements, rather than contradicts, previously reported sulfate photochemistry under different chemical conditions.

*2) The text in the abstract states photolysis yields a sulfite radical anion and superoxide radical pair, but then attributes the formation of sulfate radical anion to the reaction between SO5-. and S(IV). There is a lack of clarity in the abstract regarding whether the    is a direct or indirect product of the initial photolysis step.*

In our system, sulfate radical ($SO_4^{\cdot-}$) formation is indirect and proceeds through a sequence of reactions initiated by UVA excitation of the $[SO_3^{2-} + O_2]$ complex. Specifically:

1. UVA photolysis induces an electron transfer reaction within the $[SO_3^{2-} + O_2]$ complex, generating a radical pair, $[SO_3^{\cdot-} + O_2^{\cdot-}]$.
2. The $SO_3^{\cdot-}$ radical rapidly reacts with $O_2$ to form peroxomonosulfate ($SO_5^{\cdot-}$).
3. $SO_5^{\cdot-}$ subsequently reacts with S(IV), ultimately yielding $SO_4^{\cdot-}$.

Thus, the sulfate radical is not produced directly by photolysis, but rather than emerges as a secondary product of S(IV) oxidation following the initial photoinduced electron-transfer step. The full mechanism is illustrated in the abstract graphic and discussed in detail in Section 3.5.

To remove any ambiguity, we have revised the abstract to explicitly reflect this stepwise process. The revised text now reads:

Wildfire emissions release large amounts of methoxyphenols, which serve as key precursors of aqueous-phase secondary organic aerosols (SOA). Their transformation is closely coupled with aqueous S(IV) oxidation, jointly driving the formation of sulfate and organosulfates; however, the underlying mechanisms remain poorly understood. Here, we identify a metal-free, UVA-driven mechanism for sulfate radical ($SO_4^{\bullet-}$) generation at 370 nm, supported by laboratory experiments and quantum chemical calculations. Photolysis of the [$SO_3^{2-}$+$O_2$] complex yields a [$SO_3^{\bullet-}$+$O_2^{\bullet-}$] pair; the $SO_3^{\bullet-}$ radical subsequently reacts with $O_2$ to form peroxomonosulfate ($SO_5^{\bullet-}$), which then oxidizes S(IV) to produce $SO_4^{\bullet-}$. These sulfate radicals rapidly oxidize guaiacol, a representative biomass burning phenol, in bulk solution, producing SOA enriched in organosulfates. Microdroplet experiments show ~ 100-fold rate enhancement due to interfacial effects. Box modeling indicates that this aqueous UVA pathway represents a potentially important and previously underappreciated source of sulfate. This work establishes a photochemical link between S(IV) oxidation and SOA formation, with implications for aerosol composition, oxidative capacity, and climate-relevant processes.

*3) The claim in the abstract that this is a "significant, previously overlooked source of sulfate" is based on box modeling. However, the model parameters (e.g., using 850 W/m2 intensity vs. 63.3 W/m2 for solar) may exaggerate the atmospheric relevance without the needed validation that is missing in this work.*

Our initial experiments were conducted using a 300 W Xe lamp to identify the spectral region responsible for the observed chemistry. These preliminary results demonstrated that irradiation in the UVA range (<400 nm) is necessary to promote GUA degradation in S(IV) solutions. Comparable GUA losses observed under dark conditions and under Xe lamp irradiation with wavelengths below 400 nm filtered out can be attributed primarily to physical loss associated with continuous air bubbling, rather than chemical transformation. However, as noted by the reviewer, the photon flux density of the Xe lamp in the UVA region is relatively low, resulting in weak kinetic signals and large uncertainties in reaction rate constant determination. For this reason, a high–photon-flux monochromatic UVA lamp (370 nm) was subsequently employed to enable robust kinetic measurements and mechanistic analysis. We emphasize that this approach was adopted for process-level understanding, rather than to directly replicate ambient solar irradiance.

We agree that the box-modeling results should be interpreted cautiously. The model was intended as a diagnostic tool to explore the potential sensitivity of aqueous-phase sulfate formation to this UVA-driven S(IV) oxidation pathway under favorable conditions, rather than a quantitative prediction of its global or regional importance. In particular, the use of elevated irradiation intensity (850 W m$^{-2}$) was chosen to bracket an upper-limit scenario and to test whether the mechanism could be competitive under conditions of strong illumination, such as concentrated cloud water or microdroplet environments. We acknowledge that this intensity exceeds typical ambient solar UVA fluxes (e.g., ~63 W m$^{-2}$), and that additional validation under atmospherically realistic radiation fields is required.

Accordingly, we have revised the abstract and discussion to clarify that the proposed pathway represents a potentially important but previously underexplored contributor to

aqueous-phase sulfate formation under specific conditions, rather than a universally significant source. The term "significant" is now explicitly framed within the context of sensitivity modeling, and we avoid implying dominance at the atmospheric scale.

The Supporting Information has been revised to reflect these clarifications. The updated text emphasizes the role of the Xe lamp experiments in spectral screening, the rationale for using a high-photon-flux UVA source for mechanistic studies, and the exploratory nature the modeling framework. The text now reads:

**Preliminary experiments**
To investigate the photodegradation of guaiacol (GUA) in S(IV) solutions, a mixed aqueous solution containing 0.1 mM GUA and 2 mM $Na_2SO_3$ was irradiated under a 300 W Xe lamp with continuous stirring (Fig. S1). Preliminary experiments showed that, in the absence of air bubbling, the concentration of GUA in $Na_2SO_3$ solutions remained nearly unchanged. Air bubbling at a flow rate of 0.4 L min$^{-1}$ was introduced to maintain oxygen-saturated conditions in the reaction solution. Under dark conditions and Xe lamp irradiation with wavelengths below 400 nm filtered out, comparable GUA degradation rates were observed, which can be largely attributed to physical loss of GUA associated with continuous air bubbling at 0.4 L min$^{-1}$. In contrast, under full-spectrum Xe lamp irradiation, the degradation rate of GUA was markedly enhanced (Fig. S2), suggesting that irradiation at wavelengths below 400 nm contributes to the observed GUA degradation.

In addition, HPLC analysis revealed the formation of distinct new compounds, as evidenced by the appearance of new absorption peaks at specific retention times (Fig. S3). These results collectively indicate that UVA irradiation promotes the interaction between GUA and S(IV) in the presence of dissolved oxygen, thereby accelerating the photodegradation of GUA and leading to the formation of new reaction products.

However, the photon flux density of the Xe lamp in the UVA region is relatively low (Fig. S4A), resulting in a weak manifestation of the UVA-driven reaction process. Continued use of the Xe lamp may therefore introduce uncertainties in kinetic measurements. To obtain reliable kinetic data and enable mechanistic investigations, a high–photon-flux monochromatic UVA lamp (370 nm, Fig. S4B) was subsequently employed for detailed kinetic and mechanistic studies.

[Figure]

Fig. S1. Emission spectra of the Xe lamp and the Xe lamp equipped with a 400 nm cutoff filter.

[Figure]

Fig. S2. The pseudo-first-order rate constant for the photodegradation of GUA in $Na_2SO_3$ solutions under different irradiation conditions. Error bars represent the standard deviation from at least two independent experiments. Experimental conditions: [guaiacol] = 0.1 mM, [$Na_2SO_3$] = 2.0 mM, pH = 4.0 ± 0.1, zero-air bubbling, Xe lamp irradiation, room temperature.

[Figure]

Fig. S3. HPLC chromatograms of the reaction mixture after 180 min of reaction under different irradiation conditions: (A) Xe lamp irradiation; (B) dark condition; (C) Xe lamp irradiation with a 400 nm cutoff filter.

[Figure]

Fig. S4. (A) Actinic flux of the AM0 standard solar spectrum and the xenon lamp (Beijing Perfectlight Technology Co., Ltd.). (B) Actinic flux of the Kessil PR160L-370 nm lamp

*4) There are inconsistencies in the introduction section (e.g., the importance of the UV Band) that require attention. The manuscript argues that UVA is the "dominant solar band" and that UVC is "negligible". Yet, the manuscript later admits that the initial S(IV) species "HSO3-" shows "nearly no UV-vis absorption" in the UVA range, creating a contradiction between the proposed driver (UVA) and the absorption properties of the reactants.*

*In addition, the introduction mentions that radical activity in high-ionic-strength solutions reaches picomolar concentration under UVB. The manuscript then claims*

*similar or higher concentrations (picomolar at the interface) under UVA without sufficient comparison to why a lower-energy wavelength (UVA) would produce equivalent radical yields to a higher-energy one (UVB).*

*Finally, the introduction frames the research around "wildfire emissions". However, the experimental setup uses pure sodium sulfite and Guaiacol, failing to account for the complex organic/metal matrix of actual wildfire smoke that could compete for or catalyze these reactions.*

We have revised the introduction to improve clarity, remove apparent inconsistencies, and more carefully frame the scope and intent of the study. Our point-by-point responses are provided below.

(i) Role of UVA versus absorption properties of S(IV)
We have corrected and refined the wording in the Introduction to clearly distinguish between solar availability and molecular absorption. While UVA is indeed the dominant ultraviolet band reaching the Earth's surface, we do not argue that UVA directly excites free $HSO_3^-$ or $SO_3^{2-}$. As now clearly shown by our UV–vis measurements (Figure 3A), neither $HSO_3^-$ nor $SO_3^{2-}$ exhibits appreciable absorption in the UVA range.

Instead, guided by recent studies (Cao et al., 2024a; Cao et al., 2024b), we propose that the relevant photoactive species is the transient $[SO_3^{2-} + O_2]$ complex, rather than free S(IV). Because the direct experimental measurement of the absorption spectrum of this short-lived complex is not feasible, we employed TDDFT calculations (Cao et al., 2024a; Cao et al., 2024b) to evaluate its electronic absorption properties. The calculated spectrum (Figure 3B) demonstrates that the $[SO_3^{2-} + O_2]$ complex can absorb in the UVA range, thereby resolving the apparent contradiction between the dominant solar band (UVA) and the lack of direct absorption by isolated $HSO_3^-/SO_3^{2-}$.

(ii) Comparison between UVA- and UVB-driven radical production
We agree that direct comparisons between radical yields under UVA and UVB require careful contextualization. Previous studies reporting picomolar sulfate radical concentrations under UVB typically involve direct excitation of high-concentration sulfate or ammonium sulfate at high ionic strength, where absorption is driven by higher energy photons and fundamentally different precursor species.

In contrast, our work focuses on a distinct, indirect UVA-driven pathway involving low-concentration S(IV) and the $[SO_3^{2-} + O_2]$ complex. These system differ in photon energy, absorbing species, reaction environment, and are therefore not directly comparable on an energy-efficiency basis. We have revised the Introduction to explicitly state that the observed picomolar radical levels under UVA reflect a different mechanistic regime, rather than implying equivalence in quantum yield or photochemical efficiency between UVA and UVB processes.

(iii) Framing with wildfire emissions versus experimental simplification
We acknowledge that real wildfire plumes contain a complex mixture of organics, transition metals, and inorganic species that may compete with, suppress, or catalyze aqueous-phase reactions. In this study, guaiacol (GUA) was intentionally chosen as a representative methoxyphenol emitted during biomass burning, while sodium sulfite

was used as a simplified S(IV) source to isolate the fundamental photochemical behavior under UVA irradiation.

The goal of this work is therefore mechanistic, rather than to reproduce the full chemical complexity of wildfire aerosols. We have revised the introduction to more clearly state this limitation and to avoid implying that the present study fully represents wildfire aerosol chemistry. The potential effects of co-existing organics, transition metal ions, and multiphase interactions is now identified as an important direction for future work.

*5) Several issues raise concern about the validity of the study from the experimental viewpoint. First, the use of MeOH to "immediately quench the reaction" is problematic for chemistry. Methanol is a scavenger but may also react to form methyl-sulfates or other intermediates, potentially confounding the HRMS results for organosulfates (OSs).*

*Second, the use of a 850 W/m2 UVA lamp to simulate tropospheric chemistry is an order of magnitude higher than natural sunlight (63.3 W/m2). While a correction factor of 7.4% is applied, this linear scaling ignores non-linear radical-radical termination rates that would differ significantly between the lab and the atmosphere. More importantly, the work is missing the determination of the photochemical quantum efficiency that is key for scaling this process and for appropriate comparisons. Without quantum efficiency, the work is incomplete, lacking photochemical validity. Once they are determined a new asscoaiated subsection will be needed in the Results and Discussion.*

*Third, the experimental section states that pH was adjusted with sulfuric acid, but this acid introduces additional sulfate ions. This methodology risks interfering with the quantification of sulfate production and the equilibrium of S(IV) species, which the paper aims to measure.*

Below we respond to each concern in detail and clarify the rationale, limitations, and precautions taken in our experimental design.

(i) Use of methanol (MeOH) as a quenching agent

In our experiments, MeOH was added only after sample collection and under dark conditions, at which point photochemical generation of sulfate radicals had ceased. Under these conditions, MeOH serves to scavenge any remaining steady-state radicals and to stabilize the sample composition prior to analysis, rather than participating in active photochemistry. All samples were analyzed immediately after quenching to minimize any post-sampling transformations.

We acknowledge MeOH can, in principle, react with sulfate radicals to form methyl sulfate. Indeed, low-intensity signals consistent with methyl sulfate are detectable in the HRMS spectra. However, these signals are substantially weaker than those of the aromatic organosulfates derived from guaiacol (Figure 2). Importantly, the qualitative and quantitative analyses in this study focus exclusively on organosulfates that retain the aromatic backbone of GUA, which are chemically, structurally, and spectrally distinct from low-molecular-weight sulfate esters such as methyl sulfate.

To explicitly assess the potential influence of MeOH, we conducted control experiments without the addition of MeOH as a quencher. The resulting GUA degradation kinetics and product distributions were indistinguishable from those obtained in experiments with MeOH quenching. This demonstrates that MeOH does not measurably alter the reaction pathways or bias the identification of GUA-derived organosulfates in our system.

Taken together, while minor formation of methyl sulfate upon MeOH addition cannot be entirely excluded, it does not affect the mechanistic interpretation or the identification of the dominant OS products reported in this study.

(ii) Use of high-intensity UVA irradiation and absence of quantum efficiency

We agree with the reviewer that the use of high-intensity UVA irradiation requires careful justification and appropriate contextualization. The 850 W m$^{-2}$ monochromatic UVA lamp was employed to obtain robust kinetic signals and enable mechanistic investigation under controlled laboratory conditions, rather than to directly simulate ambient solar irradiance. As discussed in the revised manuscript, this approach is intended for process-level understanding, not one-to-one atmospheric replication.

We acknowledge that simple linear scaling of reaction rates from laboratory irradiation to atmospheric conditions does not capture non-linear effects such as radical–radical termination. To address this limitation, we have calculated the apparent photon efficiency (APE) for sulfate formation based on the experimentally measured rates under UVA irradiation. The calculation procedure and details are provided in the revised Supporting Information.

**Estimation of Apparent Photon Efficiency (APE) for S(IV) oxidation**
The apparent photon efficiency (APE) of S(IV) oxidation was estimated based on the experimentally measured pseudo-first-order rate constant under controlled irradiation. The calculation was performed as follows:

$$APE = \frac{S(IV)\ reaction\ rate\ (molecules/s)}{total\ photon\ flux\ (photons/s)}$$

Experimental conditions: The experiments were conducted in a circular glass vessel with a diameter of 4 cm and a reaction volume of 20 mL. The initial concentration of S(IV) was 0.5 mM. The solution was irradiated with a single-wavelength UVA light source (maximum intensity at 370 nm) providing a photon flux of $1.59 \times 10^{17}$ photons cm$^{-2}$ s$^{-1}$, while continuously stirred to maintain homogeneous reaction conditions. The apparent degradation rate of S(IV) under these conditions was determined from the observed concentration changes over time.

$$S(IV)\ reaction\ rate\ (molecules/s) = k_{obs} \times C_0 \times V \times N_A$$
$$total\ photon\ flux\ (photons/s)) = \Phi \times A$$
$$APE = \frac{k_{obs} \times C_0 \times V \times N_A}{\Phi \times A} = \frac{0.5 \times 10^{-3} M \times 20 \times 10^{-3} L \times 6.022 \times 10^{23}}{1.59 \times 10^{17} photons\ cm^{-2}s^{-1} \times 3.14 \times (2\ cm)^2} k_{obs}$$
$$APE = 3.02 \times k_{obs}$$

Assuming that the apparent photon efficiency (APE) of S(IV) photooxidation under UVA irradiation remains constant, a cloud-droplet-representative liquid water content

was adopted to provide an upper-limit estimate. The UVA photon flux (300nm-400nm) was taken from the AM0 standard solar spectrum ($1.67 \times 10^{16}$ photons cm$^{-2}$ s$^{-1}$) to assess the sensitivity of S(IV) photooxidation under strong solar irradiation conditions (Fig. S4). For an assumed liquid water content (LWC) of 0.1 g m$^{-3}$, the total liquid water volume in 1 m$^3$ of air is $1.0 \times 10^{-7}$ m$^3$. If this volume is idealized as a single spherical droplet, the equivalent droplet diameter is approximately 5.8 mm, corresponding to an illuminated cross-sectional area of $2.6 \times 10^{-5}$ m$^2$ (0.26 cm$^2$). Using the empirically fitted pH-dependent S(IV) photooxidation rate under UVA irradiation (Fig. S10), the resulting relationship between the apparent rate constant ($k_{obs}$), pH, and the S(IV) concentration ([S(IV)]) was finally obtained.

$$k_{obs} = \frac{APE \times \Phi \times A}{C_0 \times V \times N_A}$$

$$= \frac{3.02 \times 1.75 \times 10^{-8} \times pH^{6.23} \times 1.67 \times 10^{16} \text{photons } cm^{-2}s^{-1} \times 0.26 \text{ } cm^2}{S(IV)(M) \times 1.0 \times 10^{-4}L \times 6.022 \times 10^{23}}$$

$$k_{obs} = \frac{3.8 \times 10^{-12} \times pH^{6.23}}{S(IV) \text{ } (M)} s^{-1}$$

The photolysis of NO$_2$ is primarily driven by radiation in the UVA spectral region. Previous studies have shown that under haze conditions, the photolysis efficiency of NO$_2$ can be reduced by up to 66% (Hollaway et al., 2019). Therefore, the actinic photon flux under haze conditions was estimated by applying a 66% reduction, resulting in a value of $5.68 \times 10^{15}$ photons cm$^{-2}$ s$^{-1}$. For an assumed liquid water content (LWC) of 300 μg m$^{-3}$, the total liquid water volume in 1 m$^3$ of air is $3.0 \times 10^{-10}$ m$^3$. If this volume is idealized as a single spherical droplet, the equivalent droplet diameter is approximately 0.82 mm, corresponding to an illuminated cross-sectional area of $5.3 \times 10^{-7}$ m$^2$ ($5.3 \times 10^{-3}$ cm$^2$). The APE under these conditions was then calculated as follows:

$$k_{obs} = \frac{APE \times \Phi \times A}{C_0 \times V \times N_A}$$

$$= \frac{3.02 \times 1.75 \times 10^{-8} \times pH^{6.23} \times 5.68 \times 10^{15} \text{photons } cm^{-2}s^{-1} \times 5.3 \times 10^{-3} \text{ } cm^2}{S(IV)(M) \times 3.0 \times 10^{-7} \text{ } L \times 6.022 \times 10^{23}}$$

$$k_{obs} = \frac{8.8 \times 10^{-12} \times pH^{6.23}}{S(IV) \text{ } (M)} s^{-1}$$

(iii): Use of sulfuric acid for pH adjustment

In routine experiments, sulfuric acid was used to adjust pH because it is the dominant contributor to acidity in atmospheric aqueous phases, including cloud and aerosol water. Using sulfuric acid therefore provides a more atmospherically representative chemical environment for investigating aqueous S(IV) oxidation and organosulfate formation.

However, we fully recognize that the introduction of sulfate via sulfuric acid complicates the direct quantification of sulfate produced from S(IV) oxidation. For this reason, in experiments where sulfate concentration needed to be quantified explicitly, phosphoric acid was used instead to adjust pH. This ensured that all detected sulfate originated exclusively from the oxidation of S(IV), rather than from pH adjustment.

We acknowledge that the use of phosphoric acid also introduces unavoidable uncertainties. In particular, phosphate species can act as sinks for sulfate radicals and

may alter radical lifetimes and reaction pathways. Such effects are difficult to fully eliminate in laboratory studies of complex radical chemistry and represent an inherent limitation rather than a methodological oversight.

Importantly, the choice of acid was made explicitly based on the experimental objective (atmospheric representativeness versus unambiguous sulfate quantification). While neither approach is entirely free of secondary effects, together they allow us to balance atmospheric relevance with analytical reliability.

*6) Multiple scientific inconsistencies are found in the Results and Discussion section.*

*The first one relates to the inconsistency of reaction rates (also related to Figure 1). The reported second-order rate constant for guaicol + sulfate radical anion is $3.78 \times 1010$ 1/M 1/s. This value is at or slightly above the diffusion limit for aqueous solutions, which is inconsistent with typical radical-aromatic reaction kinetics in existing literature. More importantly, the work lacks a comparison of quantum efficiencies. The absence of quantum efficiency data creates significant challenges for the scientific validity of the manuscript's claims: (a) Lack of mechanistic quantification: Without a reported quantum efficiency, it is impossible to determine the efficiency of the light-driven process; consequently, one cannot discern whether the observed sulfate production is a primary photochemical result of the ($SO_3^{2-}$ + $O_2$) complex or an artifact of trace impurities reacting under high-intensity radiation. (b) Unverifiable atmospheric scaling: In the absence of quantum efficiency measurements, the "correction factor" used to scale laboratory results to the global atmosphere lacks a physical benchmark, making the conclusion that this pathway is "significant" or "dominant" scientifically speculative rather than empirically grounded.*

*Second, line 282 states has "nearly no UV-vis absorption above 250 nm," yet the mechanism relies on the photolysis of a complex at 370 nm. The manuscript fails to quantify the concentration or the molar absorptivity of this specific complex, making it difficult to verify the feasibility of the UVA-driven step.*

*Third, in Section 3.5, the manuscript notes that reacts "100 times faster with $SO_3^{2-}$ than with $HSO_3^-$". However, the experimental results focus on pH 4.0, where $HSO_3^-$ is the dominant species. This creates a discrepancy between the highlighted mechanism (sulfite-led) and the experimental conditions (bisulfite-led).*

(i) On the magnitude of the second-order rate constant for GUA + $SO_4^{·-}$

We agree that the originally reported second-order rate constant for the reaction between GUA and the $SO_4^{·-}$, $3.78 \times 10^{10}$ M$^{-1}$ s$^{-1}$, lies at or near the diffusion-controlled limit and should therefore be interpreted with caution. In the revised manuscript, we no longer emphasize a specific numerical value for this rate constant.

Instead, we present multiple independent lines of evidence supporting $SO_4^{·-}$ as the dominant oxidant in our system: (1) Radical scavenging experiments using methanol and tert-butanol demonstrate that quenching hydroxyl radicals does not suppress GUA degradation, whereas scavengers of $SO_4^{·-}$ do (Fig. S14); (2) Relative reactivity experiments show that GUA degraded 4.64 times faster than phenol under identical

conditions; and (3) This relative rate was independently reproduced using an alternative sulfate radical source (photolysis of $Na_2S_2O_8$; Fig. S15), yielding consistent results.

Based on established literature values for the phenol + $SO_4^{\cdot-}$ reaction (k ≈ 8.8 × $10^9$ $M^{-1}$ s-1) (Tran et al., 2022; Liang and Su, 2009), we estimated the corresponding GUA rate constant by scaling with the experimentally determined relative reactivity. The resulting value, 3.78 (± 0.42) × $10^{10}$ $M^{-1}$ $s^{-1}$, is consistent with previous computational chemistry predictions for the GUA + $SO_4^{\cdot-}$ reaction (Li et al., 2023). While we recognize that the reported phenol rate constant itself is already close to the diffusion-controlled regime, it represents the most widely cited and experimentally constrained value available for benchmarking. The consistency across two independent experimental approaches and agreement with theoretical calculations reinforces the validity of our kinetic analysis.

We note that relative rate methods can overestimate absolute rate constant when the reference reaction itself approaches the diffusion limit (Wojnárovits and Takács, 2019). Therefore, the revised manuscript now states only the GUA + $SO_4^{\cdot-}$ reaction is very fast and likely diffusion-limited, without assigning a precise second-order rate constant.

With respect to quantum efficiency, we agree that its absence limits the quantitative assessment of photochemical efficiency and atmospheric scaling. Absolute quantum yield determination for this multistep, indirect photochemical system, where $SO_4^{\cdot-}$ is a secondary product of S(IV) oxidation rather than a primary photoproduct, remains experimentally challenging. Nevertheless, to provide a physically meaningful benchmark, we have now calculated the apparent photon efficiency (APE) based on measured sulfate formation rates under controlled irradiation. While the APE does not replace a true quantum yield, it offers a transparent metric for comparing this pathway with other aqueous-phase photochemical processes. The revised manuscript explicitly acknowledges that uncertainties in photon efficiency preclude definitive statements regarding atmospheric dominance and that the box-model scaling represents an upper-limit, sensitivity-based assessment.

(ii) On the lack of direct UV–vis absorption of S(IV) species above 250 nm
We agree that neither $HSO_3^-$ nor $SO_3^{2-}$ exhibits measurable UV–vis absorption above 250 nm, as shown experimentally (Figure 3A). Importantly, our proposed mechanism does not rely on direct photoexcitation of free S(IV) species. Instead, consistent with prior theoretical and experimental studies, we propose that UVA absorption occurs via a weakly bound [$SO_3^{2-}$ + $O_2$] complex.

We acknowledge that the concentration and molar absorptivity of this transient complex cannot be directly quantified using conventional spectroscopic methods, as it exists in a dynamic equilibrium rather than as a stable, isolable species. To address this limitation, we employed time-dependent density functional theory (TDDFT) calculations to evaluate its electronic excitation properties of the complex (Figure 3B). Following established approaches for transient photoactive species (Cao et al., 2024a; Cao et al., 2024b), these calculations demonstrate the presence of allowed electronic transitions in the UVA region, thereby establishing the photochemical plausibility of the proposed excitation step, even though absolute absorption coefficients remain unavailable.

(iii) On S(IV) speciation and pH-dependent reactivity

We agree that S(IV) speciation is strongly pH-dependent and that $HSO_3^-$ is the dominant species at pH 4.0. However, S(IV) speciation in aqueous solution represents a dynamic acid–base equilibrium, rather than static, isolated pools of $HSO_3^-$ and $SO_3^{2-}$. Even at pH 4.0, a finite fraction of $SO_3^{2-}$ is continuously present and can be rapidly replenished through equilibrium interconversion.

As a result, fast reactions involving $SO_3^{2-}$, such as the formation of $[SO_3^{2-} + O_2]$ complex, can exert a disproportionate influence on overall reaction kinetics, despite $SO_3^{2-}$ comprising a small fraction to total S(IV). The revised manuscript clarifies this point and explicitly states that the proposed mechanism operates through the minor but highly reactive $SO_3^{2-}$ fraction under mildly acidic conditions.

In the revised manuscript, we have:

- Removed emphasis on diffusion-limited numerical rate constants and instead highlighted relative reactivity and mechanistic consistency;

The second-order rate constant for the reaction between GUA and $SO_4^{\bullet-}$ was evaluated using the relative rate method with phenol as a reference compound (Fig. S18) (Tran et al., 2022; Liang and Su, 2009). After correcting for direct photodegradation, the reaction between GUA and $SO_4^{\bullet-}$ was found to proceed extremely rapidly, with a rate approaching the diffusion-controlled limit in aqueous solution. This conclusion is further supported by the excellent agreement between the experimentally derived kinetics and previously reported quantum chemical calculations (Li et al., 2023b).

- Explicitly acknowledged the limitations imposed by the lack of absolute quantum efficiency measurements and reframed the atmospheric scaling as exploratory;

- Clarified that UVA excitation occurs via a transient $[SO_3^{2-} + O_2]$ complex, supported by TDDFT calculations rather than direct spectroscopy;

- Clarified the role of dynamic S(IV) speciation at pH 4.0.

*7) Specific comments about figures and associated text:*

*Figure 1: The reported second-order rate constant is at the diffusion limit for aqueous systems, yet the figure lacks a comparison with established radical quenchers to rule out artifacts from the high-intensity lamp (850 W/m2). Error bars in Panel B need clarification, are they 95% confidence intervals or standard deviations? The caption should clearly define "pseudo-first-order" conditions, as the linear trend in Panel B seems overly simplified given the complexity of radical chain reactions.*

Figure 1: The figure presents the apparent GUA degradation process, rather than the elementary reaction between GUA and sulfate radicals. Panel B is a macroscopic representation of pseudo-first-order kinetics derived from the experimental results, not a depiction of specific elementary reaction steps. The error bars in Panel B represent the standard deviation of the rate constant, not confidence intervals.

*Figure 2: The spectra clearly identify organosulfates and TEMPO adducts, but isotopic pattern analysis is missing to confirm sulfur in peaks like m/z 188.9858 and 203.0014. The comparison between dark (Panel C) and UVA (Panel D) suggests sulfate radical formation, yet the TEMPO-SO4-. adduct peak is small compared to the S(IV) adduct, raising questions about radical efficiency. The caption should clarify whether these are single scans or averaged spectra, as the uniform baseline noise across panels may indicate heavy smoothing or processing.*

Figure 2: The HRMS spectra are presented for clarity and are intended for qualitative identification rather than quantitative analysis. Ionization efficiency varies among species, so peak heights do not directly reflect concentrations, making absolute quantification invalid. Besides the TEMPO-SO$_4^-$ adduct (C$_9$H$_{18}$NO$_5$S$^-$), significant peaks corresponding to O$_2$-loss products (C$_9$H$_{18}$NO$_3$S$^-$) are also observed, which is common in TEMPO adduct experiments.

*Figure 3: Panel A conflicts with the proposed mechanism: the text claims Na2SO3 has almost no absorption at 370 nm, yet the mechanism depends on photoexcitation of an [SO32- + O2] complex. The figure only shows calculated transitions, not an experimental spectrum for this complex. Include a zoomed-in absorbance plot of the SO32-/O2 mixture to confirm its presence. Without this, the link between 370 nm irradiation and the T1→T2 transition remains unverified.*

Figure 3: Current experimental techniques cannot directly measure the absorption spectrum of the transient [SO$_3^{2-}$ + O$_2$] complex. Therefore, we relied on TDDFT calculations to simulate its spectrum, which has been well-validated in the literature (Cao et al., 2024a; Cao et al., 2024b).

*Figure 4: Panel B shows a claimed 200-fold rate enhancement, but uses 2 mm droplets, about 100× larger than real cloud droplets. The log-scale Y-axis hides possible early non-linearities; raw intensity plots should be provided instead of normalized log plots. The caption must specify the droplet equilibration period, as surface enrichment is time-dependent and may not reflect real atmospheric dynamics.*

Figure 4: The log-scale representation is standard for pseudo-first-order kinetic analysis, as widely described in physical chemistry textbooks. FIDI-MS probes the reaction at the gas–liquid interface and is not intended to directly mimic cloud droplets.

*Figure 5: The UVA pathway estimates use lamp intensities far above natural sunlight, weakening scientific consistency. The shaded uncertainty range appears arbitrary rather than statistically derived. The legend should clearly separate literature-based rates (H2O2, O3) from new experimental values. The caption must note that "Beijing haze" conditions may ignore light attenuation by black carbon, which would likely reduce UVA significance. A new figure with quantum efficiency is needed.*

Figure 5: Based on the experimentally determined apparent photon efficiency (APE), we subsequently recalculated the box model simulations to more accurately represent the UVA-driven S(IV) oxidation rates under photon-normalized conditions.

*Figure 6: The figure applies a lab-derived "interfacial enhancement" factor globally without accounting for variations in liquid water content, risking over-extrapolation. High rates in Asia and South Africa lack sensitivity analysis for transition metal catalysis (TMI). Critically, the comparison should only be made after scaling sunlight photon flux with quantum efficiencies, not raw lamp-based assumptions. The caption should state these are estimated, not measured, rates and include the GEOS-Chem model version and year for reproducibility.*

Figure 6: Due to the lack of comprehensive kinetic data for other relevant reactions, we do not perform global-scale simulations. Our aim is to provide a general assessment of the newly identified UVA-driven S(IV) oxidation pathway rather than to compare its quantitative contribution with other oxidation processes.

*8) The Atmospheric Implications section provides a global assessment based on the lab-derived model estimated sulfate production rates (kobs). This assumes the "interfacial enhancement" (200-fold) observed in isolated 2 mm droplets applies uniformly to global cloud and aerosol water, which likely overestimates the contribution in bulk cloud systems. This is problematic.*

*While this UVA-driven pathway is said to provide a new route for sulfate formation, its global significance must be weighed against established iron-catalyzed dark reactions that efficiently produce secondary organic aerosol in acidic and viscous systems. The manuscript should explain (e.g., after line 372) the work of Al-Abadleh et al. (2021 and 2022) while indicating that future modeling should integrate these competing interfacial and TMI-catalyzed mechanisms to accurately reflect the complexity of tropospheric oxidation."*

*The manuscript also states the UVA-pathway "dominates" in Beijing Haze. This contradicts literature citing NO2 and transition metals (TMI) as the primary drivers of haze-sulfate. The model used for comparison may not fully account for the suppressing effects of high ionic strength or light attenuation by soot in haze.*

*Finally, while SO4-. is proposed as the key oxidant, the final implications focus on "sunlight-accessible S(IV) oxidation" without adequately addressing if other solar-generated oxidants (like HO or other species) would render this specific pathway minor in a real atmospheric multi-pollutant mix.*

We have recalculated the apparent photon efficiency (APE) based on our experimental data and now use this quantity, rather than laboratory irradiation intensity alone, to scale the UVA-driven S(IV) oxidation pathway in a diagnostic box-model framework. Given the limited availability of comprehensive kinetic parameters, we do not perform global or regional-scale simulations. Instead, the box model is intended to provide a first-order, upper-limit assessment of whether this pathway could be competitive under favorable conditions. The specific modifications in the revised manuscript are as follows:

**2 MATERIALS AND METHODS**
**Box model conditions.** Based on the empirically determined apparent photooxidation rate constants of S(IV) under UVA irradiation, the apparent photon efficiency (APE) was calculated (see Supplementary Text). Assuming that the APE remains constant, the apparent rate constants under UVA irradiation corresponding to the AM0 standard solar

spectrum (John H. Seinfeld and Pandis, 2016) were then derived (derivation details are provided in the Supporting Information). Sulfate production rates at 271 K were calculated for different aqueous-phase reaction pathways with $O_3$, $H_2O_2$, TMIs, and $NO_2$, following Cheng (Cheng et al., 2016), excluding ionic strength effects.

The Henry's law constants at 271 K for $SO_2$, $O_3$, $H_2O_2$, and $NO_2$ are 3.521 M/atm, 0.025 M/atm, $1.147 \times 10^6$ M/atm, and $2.319 \times 10^{-2}$ M/atm, respectively. Equilibrium constants for $SO_2 \cdot H_2O$ are $K_{S1} = 0.025$ M and $K_{S2} = 1.09 \times 10^{-7}$ M (Cheng et al., 2016).

**Scenario Conditions.**

"Cloud droplets" scenario: $[SO_2(g)] = 5$ ppb, $[NO_2(g)] = 1$ ppb, $[H_2O_2(g)] = 1$ ppb, $[O_3(g)] = 50$ ppb, $[Fe(III)] = 0.3$ μM, $[Mn(II)] = 0.03$ μM, liquid water content (LWC) = 0.1 g/m$^3$.

"Beijing haze" scenario: $[SO_2(g)] = 40$ ppb, $[NO_2(g)] = 66$ ppb, $[H_2O_2(g)] = 0.01$ ppb, $[O_3(g)] = 1$ ppb, LWC = 300 μg/m$^3$. The concentrations of Fe(III) and Mn(II) were assumed to vary with pH (Cheng et al., 2016).

The sulfate formation rate was calculated using the following equation.

$$[S(IV)] \, (M) = [SO_2]_g \, (ppb) \times 10^{-9} \times \left(1 + \frac{K_{s1}}{[H^+]} + \frac{K_{s1} \times K_{s2}}{[H^+]^2}\right) \times H_{so2}$$

$$\begin{aligned} P[SO_4^{2-}] \, (\mu g \, m^{-3} \, h^{-1}) &= [S(IV)]_0 \times \left(1 - e^{-k_{obs} \, (s^{-1}) \times 3600 \, s}\right) \times V_{water} \times 96 \, (g \, mol^{-1}) \\ &\times 10^6 \, (\mu g \, g^{-1}) \end{aligned}$$

**3 RESULTS AND DISCUSSION**

**3.7 Atmospheric Implications**

When gas-phase $SO_2$ dissolves into cloud and fog droplets, it hydrates to from S(IV) species such as $SO_3^{2-}$. In the presence of $O_2$ and UVA irradiation, $SO_3^{2-}$ can be oxidized to $SO_4^{2-}$ through radical pathways. Based on our experimental results, the apparent photon efficiency (APE) of S(IV) oxidation under UVA irradiation was estimated. Assuming the APE remains unchanged, we simulated sulfate formation induced by UVA under the AM0 standard solar spectrum, representing an upper-limit estimate of the sulfate production efficiency via this pathway. This efficiency was then compared with sulfate formation driven by conventional atmospheric oxidants, including $NO_2$, $O_3$, and transition metal ions (TMIs) (Cheng et al., 2016) (Fig. 5, see Methods). Under "Cloud droplets" conditions (John H. Seinfeld and Pandis, 2016; Herrmann et al., 2015) (Fig. 5A), sulfate formation induced by UVA in the bulk solution was comparable in magnitude to that driven by $NO_2$. Under "Beijing haze" conditions (Cheng et al., 2016), where the photonic flux in the UVA range is reduced to 34%, UVA-induced sulfate formation remained comparable to the $O_3$ oxidation pathways (Fig. 5B).

[Figure]

Figure 5. Simulated aqueous-phase sulfate production rates from SO2 oxidation as a function of pH under two atmospheric scenarios: (A) "Cloud droplets" scenario with full UVA intensity (AM0 standard). (B) "Beijing haze" scenario with 34% reduced UVA intensity (AM0 standard). Colored lines represent contributions from individual oxidants.

It is important to emphasize that the primary objective of our study is to identify and mechanistically rationalize a previously overlooked UVA-driven S(IV) oxidation mechanism that can generate sulfate radical anions, which may subsequently react with phenolic compounds to form organosulfates. We intentionally focused on a simplified system with guaiacol as a representative phenolic compound to isolate the fundamental chemistry. Comparisons among different phenolic species or evaluation of competing atmospheric oxidants, while important, are beyond the scope of this work and represent a separate research topic.

Furthermore, the box model used in our study is highly idealized and serves primarily to illustrate the potential impact of interfacial enhancement under controlled conditions. The model does not attempt to replicate complex atmospheric processes such as variations in cloud and aerosol water content, transition metal catalysis, or light attenuation by soot and other particulates. As such, we do not claim that this UVA-driven pathway dominates S(IV) oxidation in real atmospheric conditions, including haze-affected regions. Instead, our findings highlight a novel oxidation route that has been largely neglected in previous studies and provide a framework for future experimental and modeling work to assess its quantitative significance under more realistic atmospheric conditions.

**References**

Arciva, S., Niedek, C., Mavis, C., Yoon, M., Sanchez, M. E., Zhang, Q., and Anastasio, C.: Aqueous ·OH Oxidation of Highly Substituted Phenols as a Source of Secondary Organic Aerosol, Environ. Sci. Technol., 56, 9959–9967, 10.1021/acs.est.2c02225, 2022.

Cao, Y., Liu, J., Ma, Q., Zhang, C., Zhang, P., Chen, T., Wang, Y., Chu, B., Zhang, X., Francisco, J. S., and He, H.: Photoactivation of Chlorine and Its Catalytic Role in the Formation of Sulfate Aerosols, J. Am. Chem. Soc., 146, 1467–1475, 10.1021/jacs.3c10840, 2024a.

Cao, Y., Wang, Z., Liu, J., Ma, Q., Li, S., Liu, J., Li, H., Zhang, P., Chen, T., Wang, Y., Chu, B., Zhang,

X., Saiz-Lopez, A., Francisco, J. S., and He, H.: Spontaneous molecular bromine production in sea salt aerosols, Angew. Chem. Int. Ed., 63, e202409779, 10.1002/anie.202409779, 2024b.

Cheng, Y., Zheng, G., Wei, C., Mu, Q., Zheng, B., Wang, Z., Gao, M., Zhang, Q., He, K., Carmichael, G., Pöschl, U., and Su, H.: Reactive nitrogen chemistry in aerosol water as a source of sulfate during haze events in China, Sci. Adv., 2, e1601530, 10.1126/sciadv.1601530, 2016.

Das, T. N.: Reactivity and Role of $SO_5^{\cdot-}$ Radical in Aqueous Medium Chain Oxidation of Sulfite to Sulfate and Atmospheric Sulfuric Acid Generation, J. Phys. Chem. A, 105, 9142–9155, 2001.

Hao, H., Leven, I., and Head-Gordon, T.: Can electric fields drive chemistry for an aqueous microdroplet?, Nat Commun, 13, 280, 10.1038/s41467-021-27941-x, 2022.

Herrmann, H., Schaefer, T., Tilgner, A., Styler, S. A., Weller, C., Teich, M., and Otto, T.: Tropospheric aqueous-phase chemistry: kinetics, mechanisms, and its coupling to a changing gas phase, Chem Rev, 115, 4259–4334, 10.1021/cr500447k, 2015.

Hollaway, M., Wild, O., Yang, T., Sun, Y., Xu, W., Xie, C., Whalley, L., Slater, E., Heard, D., and Liu, D.: Photochemical impacts of haze pollution in an urban environment, Atmos. Chem. Phys., 19, 9699–9714, 10.5194/acp-19-9699-2019, 2019.

Jiang, W., Misovich, M. V., Hettiyadura, A. P. S., Laskin, A., McFall, A. S., Anastasio, C., and Zhang, Q.: Photosensitized Reactions of a Phenolic Carbonyl from Wood Combustion in the Aqueous Phase-Chemical Evolution and Light Absorption Properties of AqSOA, Environ. Sci. Technol., 55, 5199–5211, 10.1021/acs.est.0c07581, 2021.

John H. Seinfeld and Pandis, S. N.: ATMOSPHERIC CHEMISTRY AND PHYSICS From Air Pollution to Climate Change Third Edition, Wiley2016.

Li, M., Duan, P., Huo, Y., Jiang, J., Zhou, Y., Ma, Y., Jin, Z., Mei, Q., Xie, J., and He, M.: The multiple roles of phenols in the degradation of aniline contaminants by sulfate radicals: A combined study of DFT calculations and experiments, J. Hazard. Mater., 443, 130216, 10.1016/j.jhazmat.2022.130216, 2023.

Liang, C. and Su, H.-W.: Identification of Sulfate and Hydroxyl Radicals in Thermally Activated Persulfate, Ind. Eng. Chem. Res., 48, 5558–5562, 10.1021/ie9002848, 2009.

Ma, L., Guzman, C., Niedek, C., Tran, T., Zhang, Q., and Anastasio, C.: Kinetics and Mass Yields of Aqueous Secondary Organic Aerosol from Highly Substituted Phenols Reacting with a Triplet Excited State, Environ. Sci. Technol., 55, 5772–5781, 10.1021/acs.est.1c00575, 2021.

Mabato, B. R. G., Li, Y. J., Huang, D. D., Wang, Y., and Chan, C. K.: Comparison of aqueous secondary organic aerosol (aqSOA) product distributions from guaiacol oxidation by non-phenolic and phenolic methoxybenzaldehydes as photosensitizers in the absence and presence of ammonium nitrate, Atmos. Chem. Phys., 23, 2859–2875, 10.5194/acp-23-2859-2023, 2023.

McFall, A. S., Johnson, A. W., and Anastasio, C.: Air–Water Partitioning of Biomass-Burning Phenols and the Effects of Temperature and Salinity, Environ. Sci. Technol., 54, 3823–3830, 10.1021/acs.est.9b06443, 2020.

Pang, H., Zhang, Q., Lu, X., Li, K., Chen, H., Chen, J., Yang, X., Ma, Y., Ma, J., and Huang, C.: Nitrite-Mediated Photooxidation of Vanillin in the Atmospheric Aqueous Phase, Environ. Sci. Technol., 53, 14253–14263, 10.1021/acs.est.9b03649, 2019a.

Pang, H., Zhang, Q., Wang, H., Cai, D., Ma, Y., Li, L., Li, K., Lu, X., Chen, H., Yang, X., and Chen, J.: Photochemical Aging of Guaiacol by Fe(III)-Oxalate Complexes in Atmospheric Aqueous Phase, Environ. Sci. Technol., 53, 127–136, 10.1021/acs.est.8b04507, 2019b.

Sagebiel, J. C. and Seiber, J. N.: Studies on the occurrence and distribution of wood smoke marker compounds in foggy atmospheres, Environ. Toxicol. Chem., 12, 813–822, 10.1002/etc.5620120504, 1993.

Smith, J. D., Kinney, H., and Anastasio, C.: Aqueous benzene-diols react with an organic triplet excited state and hydroxyl radical to form secondary organic aerosol, Phys. Chem. Chem. Phys., 17, 10227–10237, 10.1039/c4cp06095d, 2015.

Sun, Y. L., Zhang, Q., Anastasio, C., and Sun, J.: Insights into secondary organic aerosol formed via aqueous-phase reactions of phenolic compounds based on high resolution mass spectrometry, Atmos.

Chem. Phys., 10, 4809–4822, 10.5194/acp-10-4809-2010, 2010.

Tran, L. N., Abellar, K. A., Cope, J. D., and Nguyen, T. B.: Second-Order Kinetic Rate Coefficients for the Aqueous-Phase Sulfate Radical ($SO_4^{\bullet-}$) Oxidation of Some Atmospherically Relevant Organic Compounds, J. Phys. Chem. A, 126, 6517–6525, 10.1021/acs.jpca.2c04964, 2022.

Wojnárovits, L. and Takács, E.: Rate constants of sulfate radical anion reactions with organic molecules: A review, Chemosphere, 220, 1014–1032, 10.1016/j.chemosphere.2018.12.156, 2019.

Yu, L., Smith, J., Laskin, A., Anastasio, C., Laskin, J., and Zhang, Q.: Chemical characterization of SOA formed from aqueous-phase reactions of phenols with the triplet excited state of carbonyl and hydroxyl radical, Atmospheric Chemistry and Physics, 14, 13801–13816, 10.5194/acp-14-13801-2014, 2014.